# 'Bodhisattva Bodies': Early Twentieth Century Indian Influences on Modern Japanese Buddhist Art

## Chao Chi Chiu

Department of Art History & Archaeology, University of Maryland, College Park, MD 20742, USA; ccc024@umd.edu

**Abstract:** The first decade of the twentieth century marked a turning point for Japanese Buddhism. With the introduction of Western academia, Buddhist scholars began to uncover the history of Buddhism, and through their efforts, they discovered India as the birthplace of Buddhism. As India began to grow in importance for Japanese Buddhist circles, one unexpected area to receive the most influence was Japanese Buddhist art, especially in the representation of human figures. Some artists began to insert Indian female figures into their art, not only to add a sense of exoticism but also to experiment with novel iconographies that might modernize Buddhist art. One example included the combination of Indian and Japanese female traits to create a culturally fluid figure that highlighted the cultural connection between Japan and India. Other artists were more attracted to "Indianizing" the Buddha in paintings to create more historically authentic art, drawing references from both Indian art and observations of local people. In this paper, I highlight how developments in Buddhist studies in Japan led to a re-establishment of Indo–Japanese relationships. Furthermore, I examine how the attraction towards India for Japanese artists motivated them to travel abroad and seek inspiration to modernize Buddhist art in Japan.

**Keywords:** human figures; India; Buddhist art; inter-cultural exchange; exoticism

## 1. Introduction

When asked when modern Indian and Japanese artistic exchanges began, most scholarship points to the 1902 meeting in India between the two countries' leading cultural figures. First, from Japan, was Okakura Kakuzō (1863–1913), Japan's central promoter of Oriental art and culture and the director of Japan Visual Arts Academy (Nihon Bijutsu-in 日本美術院), one of the nation's most significant art organizations. Meeting with him was the Indian polymath Rabindranath Tagore (1861–1941), whose family was at the forefront of an Indian art revivalist movement known as the "Bengal Renaissance." Tagore also founded a prominent university named Visva-Bharati University in Santiniketan, West Bengal. The meeting between Tagore and Okakura resulted in a strong friendship between the two men and an agreement for Okakura to send students from his academy to Tagore's school to foster artistic exchanges between Japan and India. While there were many preceding cultural and religious exchanges between India and Japan, it was the meeting between Okakura and Tagore that commenced the first modern artistic exchanges between the two countries. The arrangement between the two men led to the first Japanese artists to arrive in India, paving the way for more Japanese artists to travel to the country throughout the twentieth century. Thus, the contribution of Okakura and Tagore towards initiating modern Indo–Japanese artistic exchanges holds undeniable significance.

While this colorful narrative certainly detailed the arrangements that helped the first Japanese artists to travel to India and was often promoted as an example of Indo–Japanese cultural camaraderie, the deeper story of why this meeting happened in the first place was

sometimes overlooked. The meeting between Tagore and Okakura reflected larger, more complex developments in Japanese Buddhism during the early twentieth century. Years before the two men met, Japan had been changing its perspective towards Buddhism and gaining interest in India. Buddhist clerical circles adopted academic and interpretive approaches from Western-styled Buddhist studies and Japanese scholars uncovered the history of Buddhism's origins in India and the figure of the Śākyamuni, Buddhism's historical founder.

This revolution in Buddhist studies led to the emergence of a new kind of Buddhist art that rejected Japan's centuries-old conventions by becoming more "Indianized." For example, paintings of Buddhist stories began to show figures with Indian ethnic features or attire. Female figures also appeared more prominently and seemed to embody the exoticism of India. Such provocative paintings came primarily from itinerant Japanese artists who traveled to India for inspiration, attracted by its reputation as the sacred birthplace of Buddhism. Upon their arrival, they enthusiastically studied India's ancient sculptures, visual arts, and local peoples to develop new types of Buddhist figures. These figures displayed Indian ethnicity and culture, reflecting Japan's academic shift to Buddhist teachings, a shift that emphasized the religion's Indian origins. This emphasis on Indian bodies was a signature characteristic of early twentieth-century Japanese Buddhist art.

This article traces how the development of Buddhist studies along the lines of Western academicism led to India becoming the principal reference for modern Japanese Buddhist art and an essential destination for artists seeking inspiration. It highlights how this academic shift motivated the first generation of itinerant Japanese artists to India, namely the students of Okakura's teachings, to create a new, more modern kind of Buddhist art that emphasized the religion's Indian origins. While this article focuses on Buddhist art, it also acknowledges that Japanese artists in India took inspiration from diverse sources such as their observations of local people and Hindu art. Thus, this article will also cover the creative ways that Japanese artists sought out references beyond Buddhist art to construct Indian figural types in their art.

Additionally, with an examination of the Indian-themed works produced by the first Japanese artists in India, this paper also explores the earliest instances of Buddhist art conveying the ideology of Pan-Asianism: the modern belief of Asian unity against Western Imperialism, which was fervently promoted by Okakura, and one that emphasized the spiritual unity among Asian cultures, especially between India and Japan, to act as a counterbalance against Western materialism (Jaffe 2019, pp. 17–18). In his quest for Pan-Asianism, Okakura advised Japanese artists to study Buddhist art in India, which he claimed is a crucial source that could rejuvenate Japan's traditional art. He encouraged his students to create art that embodies Asian values and artistic traditions as a counter against the influx of Western art in Japan, which he saw as threatening Japan's traditions. His students took his advice to heart because, in their works, Okakura's Pan-Asianism ideology manifested into innovative synchronizations of themes, styles, and iconographies between Indian and Japanese art as a symbolic unification between the two countries. They experimented with such synchronizations to embody Okakura's version of spiritual Pan-Asianism, highlighting the religious bond between India and Japan.

By recounting the transformations of Buddhist understanding in Japan during the early twentieth century, tracing its connection to the first Japanese artists to travel to India, and contextualizing the artworks they produced within such developments, this article lays down the foundations of Indo–Japanese artistic exchanges. It highlights the inspirations that Japanese artists looked for in India and explores recurring themes that appeared in the works they produced. The themes that early Japanese artists embodied in their Indian-themed paintings, such as notions of Indian exotic beauty and historical Buddhist paintings, will become recurring themes in the artworks of future Japanese travelers to India. Even though later artists developed new styles and iconographies, the concepts that they strove to capture shared many parallels with the interests of the first

generation. Thus, a detailed recount of the early years of Indo–Japanese artistic exchanges, including associated artists and their work, can provide us with an essential foundational understanding that can help us examine the works of all Japanese artists who traveled to India throughout the twentieth century.

## 2. Early 20th Century Buddhist Studies in Japan: Discovering the Indian Connection

As previously mentioned, the years leading up to Okakura and Tagore's meeting (from the late nineteenth to the early twentieth century) were classified by major changes in Japanese understanding of Buddhism and its relationship with India. While many Japanese people had known that Buddhism originated in India ever since pre-modern times, few displayed an accurate understanding of the place. Commonly referred to as *Tenjiku* (天竺), India had always been portrayed as a mythical land that was seldom visited by Japanese Buddhists outside of stories and dream-like visions (Licha 2021, p. 330). Furthermore, while most Japanese knew Buddhism originated from India, not many possessed a precise comprehension of the differences in Buddhist practices between India and Japan. To begin with, Indian Buddhist traditions lie within what contemporary scholars identify as Theravada Buddhism in current Buddhist nomenclature: the original form of Buddhism founded in ancient India and characterized by its adherence to the teachings of Śākyamuni, the historical Buddha and founder of the religion.

In contrast to India, Buddhist practices in Japan fall under the overarching category of Mahāyāna Buddhism, which is commonly framed as the counterpart to Theravada Buddhism. By the late nineteenth century, Japanese Buddhist scholars have also begun to use the term "Eastern Buddhism" to distinguish their religious practices from Indian traditions (Snodgrass 2012, p. 83). Unlike Indian Buddhism, East Asian Buddhism deified the Buddha and portrayed the figure as a divine cosmic god rather than a historical spiritual leader. Additionally, in medieval and pre-modern Japan, Buddhist devotees worshipped a pantheon of deities and Bodhisattvas in addition to the historical Śākyamuni (Auerback 2016, pp. 3–4). Admittedly, in Japanese Buddhist practices, the figure of Śākyamuni does appear among the many incarnations of Buddha and other deities as an icon of worship and devotion. However, his historical identity tends to be downplayed in favor of a more divine representation (Auerback 2016, p. 4). This is most evident in the theory of *trikāya* or "three bodies" from Mahayana Buddhist doctrines, which proposes that the Buddha possesses three distinct incarnations or bodies: his *dharmakāya* or "body of truth," his *sambhogakāya* or "body of reward," and finally his *nirmānakāya* or "body of transformation." Continually, Śākyamuni is commonly associated with the "body of transformation," *nirmānakāya*, where the Buddha physically manifested on this world as the historical founder of Buddhism to guide sentient beings (Auerback 2016, p. 13).[1] As the theory of *trikāya* showed, while Japanese devotees acknowledged the existence of Śākyamuni, they perceived him as simply the physical representation of the Buddha, lesser in importance to other more metaphysical and divine forms.

While this has been the conventional portrayal of Śākyamuni in Japanese traditions leading up the modern period, Japanese Buddhist scholars gained a renewed interest in the historical Buddha as it transitioned to the late-nineteenth century due to exposure to European scholarship on Buddhism. As Japan underwent intense modernization as it entered the Meiji period (1869–1912), opportunities for overseas travel opened up and Japanese scholars were able to study Buddhism according to Western scholarly methodologies in Europe. Notable individuals included Nanjō Bunyū (1849–1927) and Kitabatake Dōryū (1820–1907). Both traveled to London and studied under the philologist Max Müller (1823–1900), one of the leading figures in oriental and Buddhist studies at the time (Harding 2008, p. 30). Under the influence of Western scholarship, Japanese scholars and artists began to apply a scientific and historical approach to studying Buddhism. Continually, Western publications from Europe and America on the life of Buddha were also translated and distributed across Japan by the 1890s, which became an attractive topic

among Japan's intellectual and artistic circles (Auerback 2016, pp. 165–66). Examples include the 1894 book *The Gospel of Buddha* by Paul Carus (1852–1919), which was translated into Japanese by religious scholar Suzuki Daisetsu (also known as D. T. Suzuki, 1870–1966) and read widely by the Japanese public (Ibid, p. 204). Through interactions with Western scholarly knowledge and distributed publications, Japanese Buddhist intellectuals began to change their understanding of Buddhism with an affirmation of India as the birthplace of Buddhism and a renewed interest in the character of Śākyamuni.

Furthermore, under Western scholarship's influence, India rapidly garnered interest from Japanese Buddhists and many designated it as an attractive, even essential, destination for Buddhist pilgrims to visit. With the possibility of international travel by the turn of the twentieth century, India went from an imagined land to a crucial destination in the pursuit of Buddhist knowledge for Japanese priests and intellectuals. For example, on his way back to Japan from Europe, Kitabatake visited Buddhist archaeological sites across India associated with the life of Śākyamuni (Jaffe 2019, pp. 29–31). Most notably, Shaku Sōen (1860–1919) traveled to Ceylon (modern-day Sri Lanka) to study the precepts of Theravada Buddhism and toured religious sites in India (Harding 2008, pp. 66–67). The journeys and activities of these early Japanese explorers in India would later become the archetypal pilgrimage for future Japanese travelers, including artists, to follow.[2]

Yet, while Japanese people's interaction with Western scholarship ignited their interest in India and its Buddhist traditions, it also brought about competition. As Stephen Kigensan Licha described, many Japanese Buddhist scholars objected to Western academia's presentation of Indian Buddhism as the more "pure" Buddhism over Japan's Mahāyāna Buddhist traditions, which Westerners deemed as superstitious (Licha 2021, p. 330). Fighting against Western scholarship's dismissive attitude towards their practices, Japanese Buddhist apologists used various strategies to fashion their Buddhist traditions as a modern religion that is more philosophical and rationalized (Snodgrass 2012, p. 83). For example, as Richard Jaffe observed, published accounts of Kitabatake's journey to Indian Buddhist sites presented a narrative of Indians and their Buddhist practices as primitive, requiring the intervention of itinerant Japanese Buddhist clerics like Kitabatake to bring to Japan where it can be successfully modernized (Jaffe 2019, pp. 30–31). As such narrative shows, Japanese Buddhist apologists strategically framed India and its Buddhist practices as inferior to put Japanese Buddhist practices in a better light. Many regarded India as an Asian country humiliatingly subjugated by European powers, and they associated Indian Buddhism with a "lesser" form of Buddhism practiced by a colonized population (Licha 2021, pp. 330–31). While Japanese Buddhists acknowledged the scholarly and religious significance of India, they also perceived it as a competitor for international recognition.

Japan's mixed perception towards India was also prominently displayed at the World Parliament of Religions, a conference of world religions held at the World Columbian Exposition in Chicago in 1893, which played a crucial role in establishing India's importance in Japanese Buddhist Studies. Additionally, it was at this event that scholars from around the world delineated the framework for Buddhist nomenclature. Many of the aforementioned terminologies such as Theravada, Mahāyāna, and other Buddhist classifications underwent clarification. One of the invitees at the Parliament was the Sri Lankan Buddhist leader Anagarika Dharmapala (1864–1933) as the representative of Theravada Buddhism, and his participation helped India be recognized globally as the birthplace of Buddhism. At the same event, Japanese delegates also attended to represent their version of Buddhism, which laid a firm foundation for the dichotomy between Mahāyana (Japanese) and Theravada (Indian) Buddhism (Ibid, p. 332). However, while the World Parliament of Religions helped incite global interest in India and differentiated between Japanese and Indian Buddhism, it also allowed the Japanese Buddhist delegates to subtly present a Buddhist hierarchical structure that framed Mahāyāna Buddhism as superior to Indian Buddhism. For example, attendants at the congregation gave Indian

Buddhism the label of Hīnayāna Buddhism, meaning the "low vehicle" acting as the counterpart to Mahāyana Buddhism, the "greater vehicle." Such verbal strategies acted as another example of Japanese Buddhists framing their practices as superior.

From a fascination with India and Śākyamuni to framing Theravada Buddhism as inferior, there were a range of perceptions towards India and its version of Buddhism among early Japanese Buddhist scholars and apologists, and their perspectives will have some parallels with Japanese Buddhist artists' perception of the subcontinent following the World Parliament of Religion. Even though many respected India as an authority on Buddhism, many also stereotyped the country and its people as an exotic "Other" in paintings. For example, some painters portrayed Indian people, especially females, as dark-skinned and scantily dressed. Others featured India's wild tropical vegetation and animals as backdrops in their works, creating a romanticized picture of a foreign and ancient land that was technologically inferior but spiritually superior to Japan.

**3. Effect on Modern Japanese Art—Okakura Kakuzō and "New Buddhist Art"**

As Japanese scholars intensified their focus on the history of Buddhism in Japan and the figure of Śākyamuni, the transformation naturally affected Japan's artistic circles, which saw great potential for new subject matters in the narrative of Śākyamuni's biography. Blurring the lines between religious and historical paintings, artists began to depict episodes from Buddha's life and experimented with new visual strategies that presented Śākyamuni as an Indian man, rather than the cosmic deity that was worshipped for centuries in Japan. Furthermore, the individual who most fervently transferred such developments to the art world was Okakura Kakuzō. Okakura was one of the most important figures in modern Japanese art history, known for his endeavors in defending Japan's artistic traditions as the nation rapidly modernized. He was the director of Japan's prestigious Tokyo University of the Arts (Tōkyō Geijutsu Daigaku 東京芸術大学) from 1887 until his resignation in 1898. Bringing several of his closest students with him, Okakura founded the competing Japan Visual Arts Academy, dedicated to the study of Japanese-style painting, better known as *nihonga* (日本画). Thus, many of Okakura's students went on to become renowned *nihonga* artists, such as Yokoyama Taikan (1868–1958), Hishida Shunsō (1874–1911), and Shimomura Kanzan (1873–1930).

In modern Japanese art history, *nihonga* is often presented as the counterpart to Japanese Western-style painting, or *yōga* (洋画), and used to describe a diverse array of artworks that utilized traditional Japanese themes, materials, and techniques. However, this does not indicate that *nihonga* artists are strict followers of conventions. *Nihonga* artists under Okakura often clashed with conservative artists due to their eccentric techniques and themes in their attempt to "modernize" Japanese painting. One example is the creation of the *mōrōtai* (朦朧体), which is best represented by Yokoyama Taikan's *Towing a Boat* (Figure 1). Usually translated as "vague" or "hazy" style, *morotai* included a lack of outlines and strong contrasts between light and dark areas which creates a foggy ambiance, hence its name (Inaga 2009, p. 152). While the style was not well-received in Japan, it was better received by Indian artists when Taikan and Shunsō introduced it during their time abroad, further encouraging artistic exchanges between Indian and Japanese artists (Satō 1998, pp. 78–79).

Apart from encouraging his fellow artists to try new styles and techniques, Okakura emphasized the importance of Buddhist art in modernizing *nihonga*. With India garnering worldwide interest as the birthplace of Buddhism following the World Parliament of Religion, Okakura encouraged his students to travel to India to rediscover their nation's spiritual roots. More specifically, he wanted Japanese artists to study India's ancient Buddhist arts, which he claimed were the predecessors to Japan's religious art (Inaga 2009, p. 152). Okakura showcased his dedication to Buddhist art when, in 1895, he organized a painting competition where artists were charged to depict episodes from the life of Buddha "as contained in *The Gospel of Buddha* written by [Paul] Carus." Among the set of rules he drew for the competition, Okakura emphasized that his competition aimed to

"widen the scope [of Buddhism], and to give a new stimulus to Buddhist art." He expressed his discontent with traditional Buddhist art in Japan, which he believed fell short in capturing the vibrant and dynamic aspects of the Buddha's life with its adherence to repetitive iconographies for centuries.[3] For instance, Okakura described how "previous Buddhist images (*butsuzō* 仏像) have all copied the single posture of the Buddha in meditation (*zenjō* 禅定)" with little variety. In contrast, he described how European Christian art vividly portrayed episodes from the life of Jesus with diverse iconographies and expressions.[4]

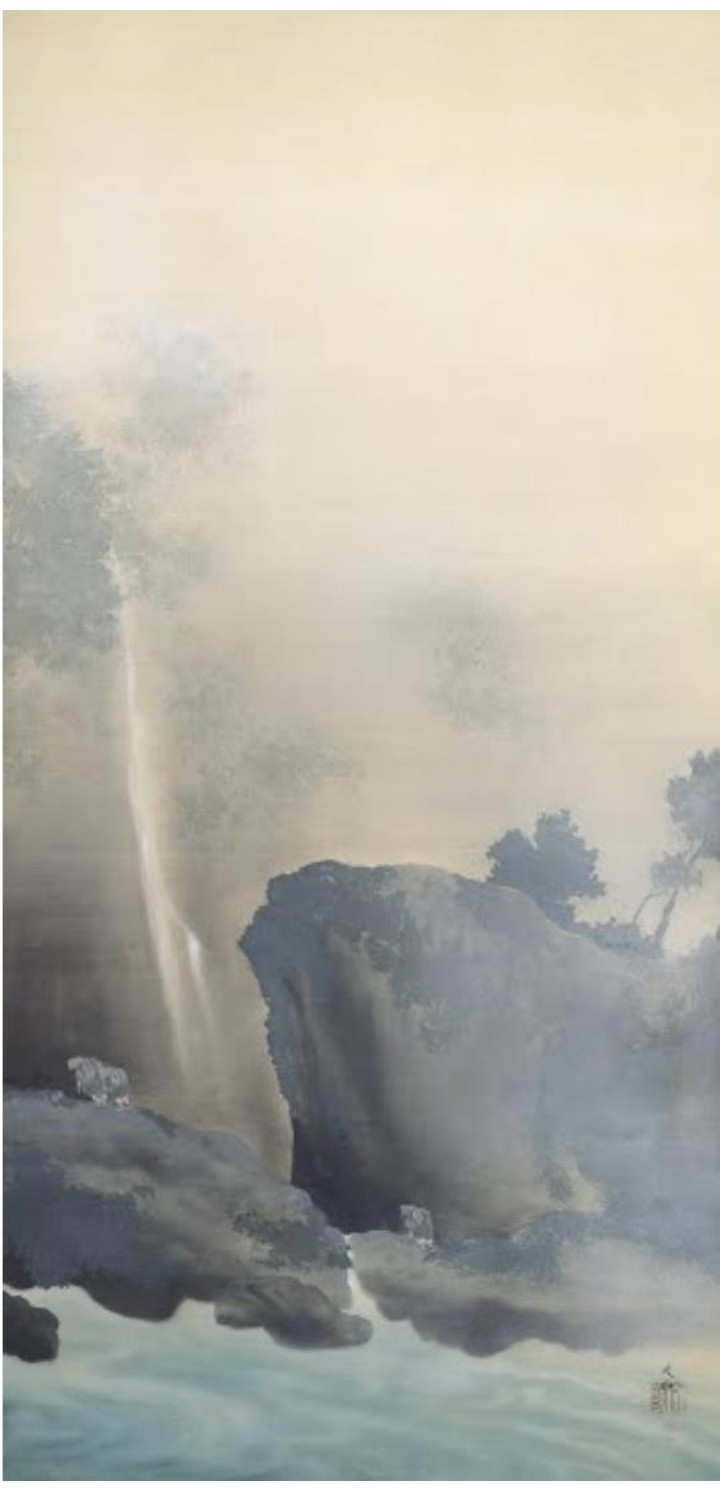

**Figure 1.** Yokoyama Taikan 横山大観, *Towing a Boat* (*Hikifune* 曳船), 1901, ink and color on silk, 27 x 54 in. (68.6 x 139.4 cm). Adachi Museum of Art. Image provided by the museum.

Among the first to respond to Okakura's call for new Buddhist paintings was his colleague, Shimomura Kanzan, with his scroll painting, *Buddha's Birth* (Figure 2), which was submitted to the Autumn Exhibition of the Japan Painting Association (Nihon Kaiga Kyōkai 日本絵画協会) in 1896. The painting depicts a newborn Śākyamuni standing upright on a lotus surrounded by figures in noteworthy outfits. To the left of Śākyamuni stands a figure with his chest exposed with only a single cloth draped over his torso. Meanwhile, the figure to Śākyamuni's right is draped in a cloth that resembles Persian textiles found at Japan's Shō-sōin (正倉院) repository with their iconic blue medallion designs (Ogata 2012, pp. 2–5). Together with the figure with the exposed chest, Kanzan depicted two figures in clothing that invoked Japanese imaginations of ancient civilizations from the far West, highlighting the foreign exotic origins of Buddhism. When the painting was displayed, some critics immediately pointed out the Indian influences apparent in the outfits of the retainers. One report stated how Kanzan "took into account the dress and customs of India, by the recent development of archaeology," while criticizing it for "slightly lacking in elegance."[5] Another review expressed uncertainty regarding the painting's Indian influence, stating how "the characters differ from previous Buddhist paintings, but they can't be regarded as definitely Indian, either," followed by a scathing remark saying that the piece is "lacking in vitality."[6] These reviews demonstrated disparate opinions of Japanese viewers on the adoption of Indian elements in Buddhist paintings. Some disagreed about the appropriate attire to represent Indian people and many found Kanzan's Buddhist figures unconventional and difficult to judge. Nevertheless, Kanzan's painting presented an example of artists answering Okakura's call to rejuvenate Buddhist art by adapting Indian elements.

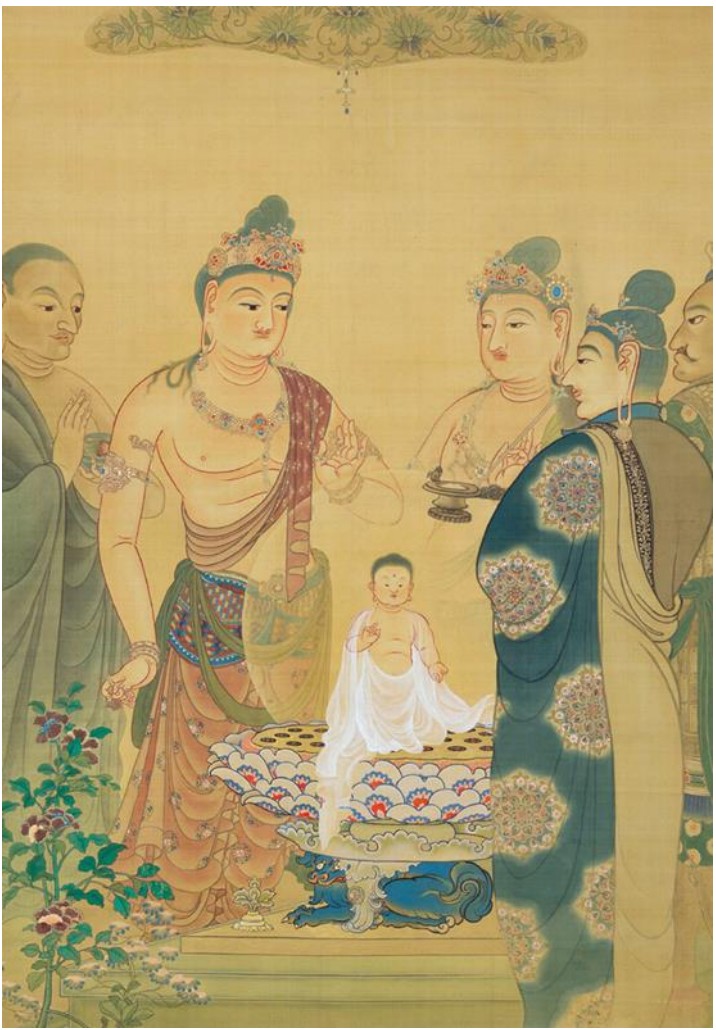

**Figure 2.** Shimomura Kanzan 下村観山, *Buddha's Birth* 仏誕, 1896, ink and color on silk, 79.9 x 56.5 in. (203 x 143.5 cm). Tokyo University of the Arts. Image provided by the university.

Apart from his endeavors in encouraging new forms of Buddhist art, Okakura also called for stronger cultural and spiritual unity between Asian countries in the face of Western Imperialism, which became the backbone of his ideology of Pan-Asianism, outlined in the publication of his famous English-language book *The Ideals of the East.* While Okakura is remembered today as an advocate for Pan-Asianism, most people sadly overlooked the contributions that India had in the formation of his ideology. At the time, many Indian intellectuals shared Okakura's Pan-Asianist thinking. Most notable was the Hindu reformer Swami Vivekananda (1863–1902), whose impassioned speech on spiritual universalism and harmony at the 1893 Parliament of World Religions resonated greatly with Okakura. In fact, Okakura traveled to India in 1902 specifically to invite Vivekananda to lecture in Japan. During his travel, he became acquainted with other intellectuals who shared his belief in Pan-Asianism, such as Sister Nivedita (1867–1911), an Irish expatriate and crucial figure in Indian nationalism, and the aforementioned Rabindranath Tagore. Sister Nivedita even helped translate several of Okakura's Japanese lectures and collaborated with him in writing *The Ideals of the East* when he was in India (Inaga 2004, p. 130). Thus, we can see how Okakura's time abroad contributed to his Pan-Asian perspective.

The examples above illustrate how Okakura connected many contemporary developments to Japanese Buddhism and Indo–Japanese relationships to the art world. Following Japan's new trend of focusing on Buddhism's history, he encouraged his artists to draw inspiration from the life of Śākyamuni. Working with contemporary Indian intellectuals, he developed his ideas for Pan-Asianism. Among the Indians with whom he worked, Tagore was the most significant in initiating artistic exchanges. Tagore suggested that Okakura send his artists to his school to initiate a series of exchanges where Okakura's students could teach Japanese art techniques to his family. Thus, in 1903, Okakura advised two of his brightest students to travel to India: Yokoyama Taikan and Hishida Shunsō (Wattles 1996, p. 51).

## 4. Taikan and Shunsō in India—Visualizing the Indian Female in Buddhist Art

Today, Yokoyama Taikan and Hishida Shunsō, along with the aforementioned Shimomura Kanzan, are considered among the earliest and most crucial contributors to the genre of *nihonga*. Their artworks reflected the principles of their mentor Okakura: to restore appreciation for Japan's traditional arts, explore possibilities to modernize Japanese-style paintings, and preserve Asian values against the encroachment of Western art and culture. Yet, despite being the most respected *nihonga* artists today, Taikan and Shunsō's "Indian period," lasting between 1903 and 1909, received little attention, even though it was a formative period for both artists and has presented us with several important examples of their early-career works. Their travels to India in 1903, under the encouragement of Okakura, defined an important chapter in both artists' careers.

Even though Okakura instructed Taikan and Shunsō to teach the Tagore family Japanese painting techniques and study India's Buddhist art, the two young students' first paintings in India took more inspirations from Hindu rather than Buddhist themes. During their time at Tagore's school, the family commissioned both artists to depict traditional Indian stories to observe how Japanese art techniques could be applied to Indian themes. One member who interacted closely with Taikan and Shunsō was Abanindranath Tagore (1871–1951), the nephew of Rabindranath Tagore and an important figure in Indian modern art. While the commissioned paintings by Abanindranath and others were not Buddhist, they gave the Japanese artists their first experience in depicting Indian figures in their paintings. These early paintings also received reviews in the 1922 article *Indo-Japanese Painting*, published in the popular Indian art periodical *Rupam* by an anonymous author (Gangoly 1922). Despite the long period between its publication and

Taikan and Shunsō's stay in India, the article provides us with an insightful look at the reception of both artists' early paintings by Indian audiences. Among the earliest paintings Taikan produced in India are *Ras Lila* (Figure 3) and *Indian Guardian* (*Indo Shūgo-Jin*, Figure 4).

*Ras Lila* took the form of a silk painting that Taikan painted per a request by Abanindranath Tagore to decorate one of his atelier's walls with a large painting in 1903. The title refers to a well-known story about the Hindu deity Krishna dancing with his wife Radha and several milkmaids (*Gopīs*), signifying the love between a divine being and mortals. Furthermore, "Ras Lila" has been a popular motif in Indian art since the 16th century, ranging from Rajput paintings to Mughal miniatures. While there were several variations in how the subject matter was depicted in Indian art, one consistent iconography is the crowd of females dressed in elaborate outfits dancing around the figure of Krishna. By asking Taikan to paint this well-known Indian tale, Abanindranath eagerly wanted to see what innovation Taikan could bring to a traditional Indian motif like Ras Lila through the medium of Japanese art (Satō 2002, pp. 3–5).

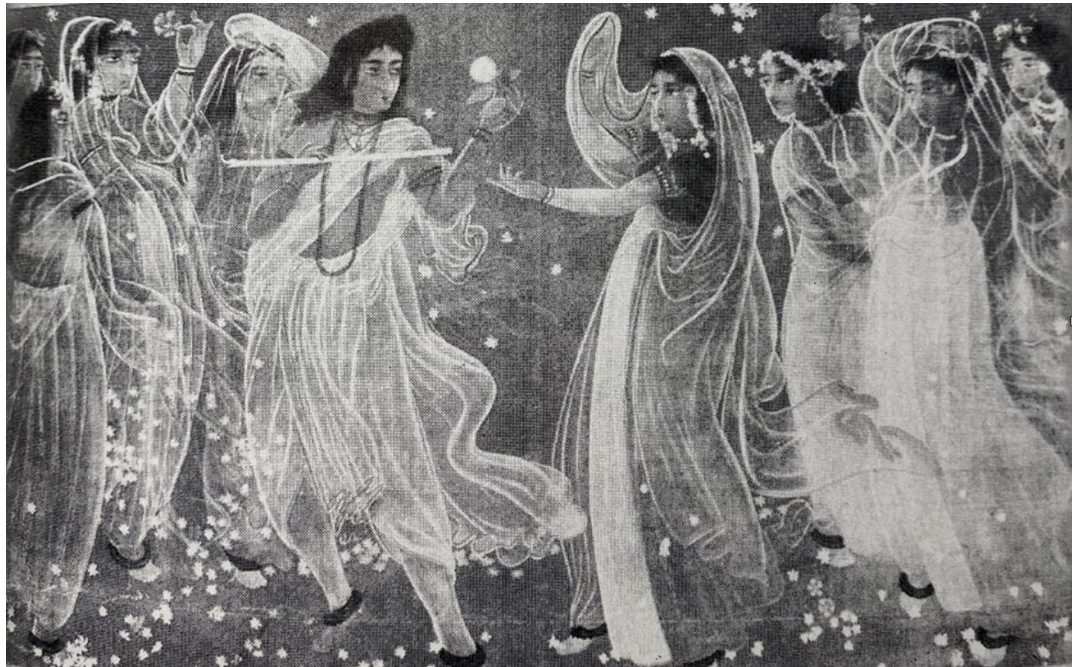

**Figure 3.** Yokoyama Taikan, *Ras Lila*, 1903, Photo Reproduction, dimensions unknown. Image from "Indo-Japanese Painting," *Rupam* . Image: Public Domain.

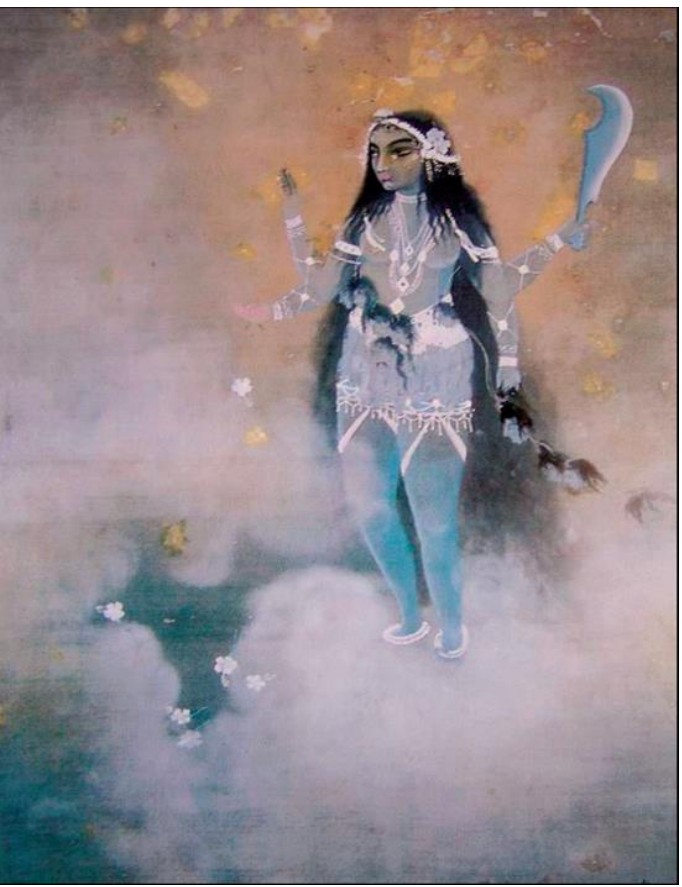

**Figure 4.** Yokoyama Taikan, *Indian Guardian (Indo shugoshin* 印度守護神), 1903, ink and color on silk, 27.6 x 20 in. (7 x 51 cm). Image from *Yokoyama Taikan zuroku (*横山大観図録) published by Dai Nihon Kaiga. Image: Public Domain.

Curiously, after he was given a rudimentary explanation of the story of Ras Lila, Taikan did not ask to look at Indian art for references. Instead, as recorded in Abanindranth's famous memoir *Jorasankor Dhare* (*On the banks of Jorasankor*), Taikan asked to see how Indian women wear the *sari* (traditional Indian garment for women). He requested Abanindranath's daughter to model for him while wearing the garment. In addition to a live model, Taikan also consulted photographs of archaic art and sculptures to better understand Indian garments and the way they were worn, doing ample research to ensure an authentic portrayal of Indian females (Satō 2002, 3-5). Unfortunately, no visual records remain of Yokohama's painting save for a photographic reproduction that appeared as the frontispiece of *Indo-Japanese Paintings*. However, even through a black-and-white reproduction, viewers can see the dexterity of Taikan's hand. Leaving few empty spaces, Taikan filled the painting with dancing figures whose bodies, from head to toe, take up the entire vertical space of the painting. The *sari* and outer garments of the women and Krishna also appear diaphanous and suggest movement through their flowing motion.

In the same year, Taikan painted *Indian Guardian*, which depicted the multi-armed, blue-skinned Hindu goddess, Kali. At the time of Taikan's visit, the image of Kali had become a widely distributed icon in India through lithography, with the most famous example being an image used to advertise cigarettes (Figure 5) (Mitter 1994, pp. 210–15). Thus, Taikan had no shortage of contemporary artistic examples to use as references. In Taikan's configuration of Kali, he displayed all the conventional motifs associated with the Hindu goddess, including the multiple limbs, the blue skin, and the necklace of human heads. However, if we compare Taikan's version of Kali to contemporary Indian

portrayals of the goddess, such as the cigarette advertisement, we can see how his portrayal diverged drastically from the popular image that was widespread in India.

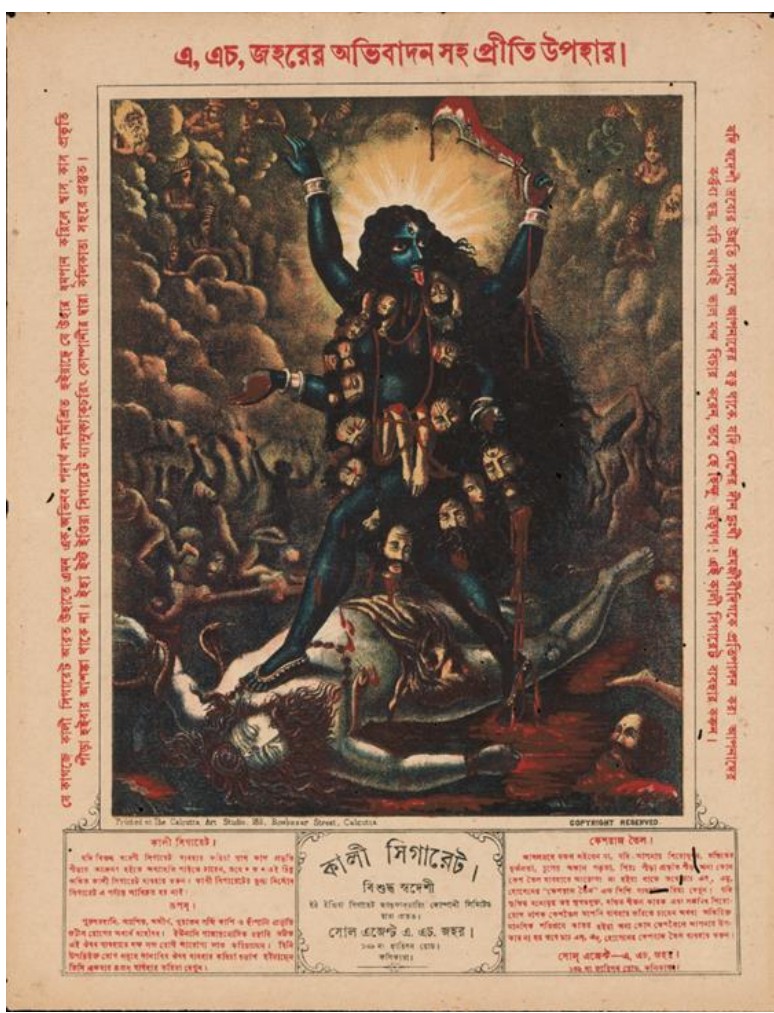

**Figure 5.** *Kali Lithograph*, 1885–95, lithograph, 17 x 13 in. (43 x 33 cm). The Metropolitan Museum of Art. Image provided by the museum. Purchase, Anonymous Gift, 2013, https://www.metmuseum.org/art/collection/search/78257.

Using popular cigarette advertisements as an example, the figure of Kali contained several examples of violent imagery, such as blood dripping off the goddess' cleaver and her necklace of severed heads. The goddess also appeared more belligerent with her wide stride, foot on top of a man, and her tongue sticking out. In contrast, Taikan's Kali figure appeared more subdued without any blood while standing still with a serene expression. The unconventional way that Taikan portrayed Kali was not lost on Indian locals, as evidenced by the scathing review his work received in *Indo-Japanese Paintings*:

> In the benign and even, charming face, the artist has forgotten the original conception of "The Terrible One" (Kali), familiar to us in the Indian versions […] Our Japanese illustrator of "Kali" has [substituted] an attractive face in place of the traditional "repulsive" one, incidentally eliminating the feeling of the terrible (the bhayânaka rasa), which is the staple part of the conception. (Gangoly 1922, p. 41)

As the review showed, by toning down the macabre and combative aesthetics associated with Kali, Taikan's painting failed to capture the goddess's iconic characteristics.

In the same article, Taikan's *Ras Lila* received similar criticisms for failing to capture the artistic conventions of Ras Lila in Indian art. One section of the article claimed that:

> It was useless to claim that the artist has been able to adequately picture the personality of Krishna, notwithstanding the flourish of the flute and Kadama flowers. The "Gopis," the dancing milkmaids, are presented without any respect for, or understanding of, Indian types. (Gangoly 1922, p. 42)

These criticisms toward Taikan's two early paintings from *Indo-Japanese Paintings* revealed how rather than learning its conventions, Taikan utilized Indian art more like a vessel for his fantasies of India. His preference for a live female model over contemporary Indian works as a reference for *Ras Lila* and his omission of the threatening traits of Kali all showed a lack of interest in staying faithful to Indian artistic conventions. Instead, Taikan treated Indian figures more like a platform where he could construct his ideal and fantasized female figures, or as a vessel to display his study of Indian women.

Around the same time that Taikan produced his two paintings, Shunsō also experimented with Indian-themed paintings, evidenced by his depiction of the Indo–Japanese Buddhist deity in *Benzaiten* (Figure 6), a Buddhist goddess that came from East Asian interpretations of the Hindu goddess of learning, Sarasvati. By choosing a goddess that is simultaneously Japanese and Indian, Shunsō depicted the spiritual and cultural unity between the two Asian countries. Additionally, not unlike Taikan's *Indian Guardian*, Shunsō's Benzaiten figure took inspiration from widely circulated images of Sarasvati in India, such as a popular lithograph produced by Calcutta Art Studio (Figure 7). Unlike Taikan, however, Shunsō directly molded his figure after the lithograph image, with his Benzaiten having the same pose as Sarasvati from the lithograph (Satō 2010, pp. 3–4). Furthermore, he also incorporated various signs from the lithograph into his painting to make his figure unmistakably Indian. For example, the goddess holds an Indian *vina* (a lute-like instrument) and wears a *sari* with jewelry adornments. While Shunsō's figure mimicked the adornments and pose of the lithograph's Sarasvati, its face and clothing diverged from it. Instead of the large eyes of the lithograph figure, Shunsō depicted Benzaiten with slender eyes, giving her an appearance more Japanese than Indian. He also painted his figure in simple white translucent cloths rather than the colorful red and orange attire worn by the lithograph figure. While it is unclear if Shunsō purposefully depicted Benzaiten with Japanese features or simply used a familiar style, his painting revealed how while he referenced Indian images more than Taikan, he also could not help but apply his personal touches just like his colleague.

From the early works of Shunsō and Taikan, we can see how Indian female bodies captivated both artists. Taikan requested Abanindranath's daughter to model for him in a *sari* and, as the reviews in *Indo-Japanese Painting* noted, he hardly followed Indian artistic traditions. Instead, Taikan focused more on capturing the bodies of Indian females rather than following traditional female iconographies in Indian art. Meanwhile, Shunsō was more open to looking at contemporary images of Indian female figures for reference and following their conventions. However, even he dispensed with Indian traditions to make his goddess figure appear more Japanese. While their approaches differed, both Taikan and Shunsō focused on Indian female bodies in their early works.

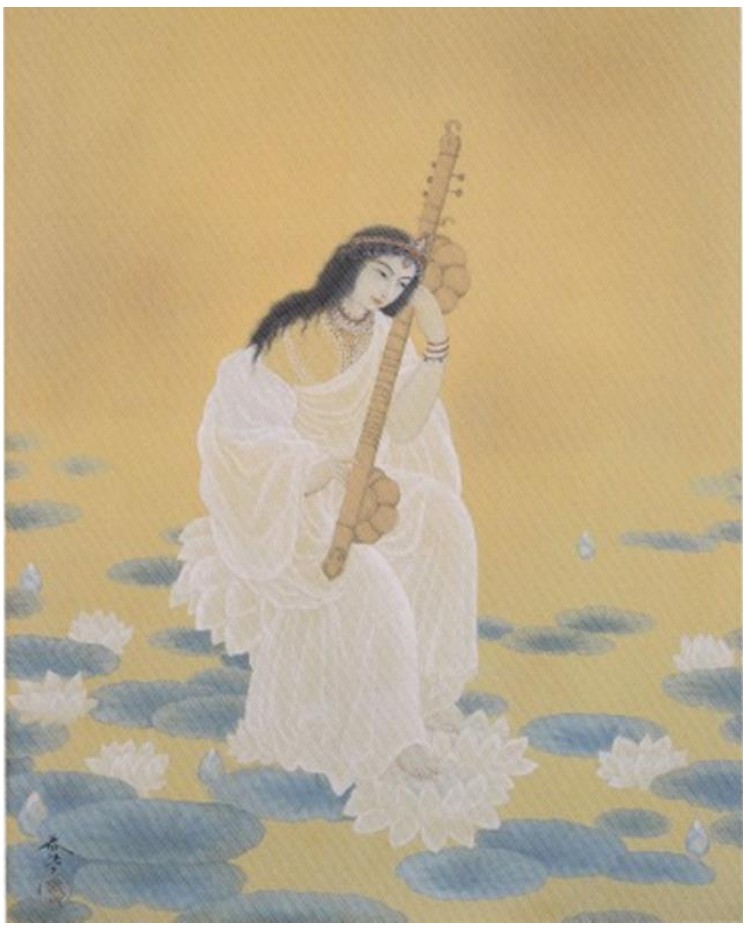

**Figure 6.** Hishida Shunsō 菱田春草, *Benzaiten* (弁財天), 1903, ink and color on silk, 15.8 x 19.7 in. (40 x 50 cm). Private Collection. Image: Courtesy of the owner.

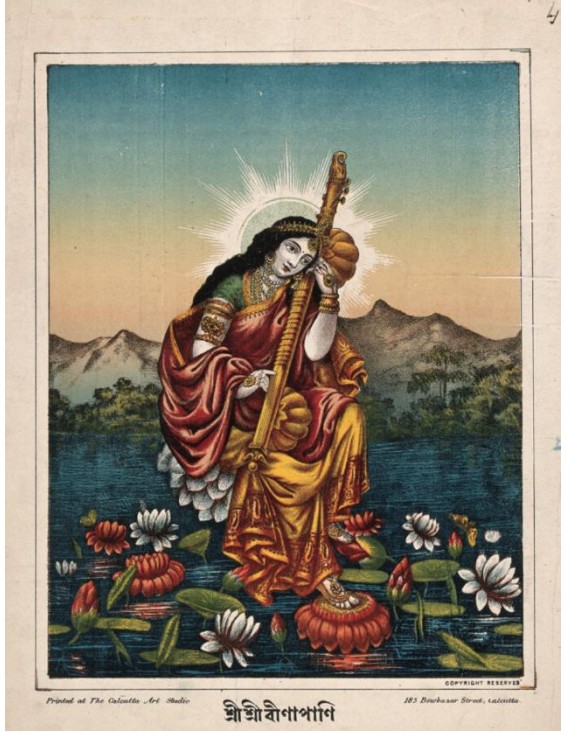

**Figure 7.** Calcutta Art Studio, *Sarasvati,* Late 19th century, lithograph 11.8 x 9.3 in. (30 x 23.7 cm). Welcome Collection, London. Image: Public Domain.

On the surface, it appears that both artists' attraction to Indian women revealed their objectification of India and its people. While it is undeniable that the two young artists were drawn by the allure of an exotic land and acted on their fantasies of India, a closer look at the historical context of both paintings shows more complex motivations behind their attraction to Indian female bodies. To begin with, Taikan and Shunsō's paintings of Kali and Sarasvati (Benzaiten) were produced in response to an Indian woman's request: Abanindranath's niece, Sarala Devi Chaudhurani (1872–1945), one of India's most notable feminists who contributed immensely to women's rights and education (Togawa 2023, p. 62). As recorded in her autobiography *Jibaner Jhalapara* (*The Fallen Leaves of Life*), Chaudhurani requested the Japanese artists to produce paintings of Indian goddesses because, similar to Abanindranath, she wished to see how Japanese art could introduce new icons and styles to the Bengal Renaissance (Togawa, p. 44). Thus, more than just portraying Indian women to satisfy their exotic fantasies, Taikan and Shunsō collaborated with contemporary Indians to create a female icon: a trans-culturally fluid "Oriental Woman" that united all Asian cultures.

By the time Okakura sent his students to India, Indian female figures had been gaining prominence as symbols of Indian pride and rebellion against Western imperialism, usually as personifications of India or Bengal as a "mother country." This popular representation of India as a maternal nationalist icon can be traced back to a 1873 poem titled *Vande Mataram* (*I praise you*) by Bengali novelist and poet Bankim Chandra Chatterjee. In his poem, Chatterjee personified Bengal as a "mother goddess," and the iconography became a popular nationalist symbol for Bengali intellectuals such as Rabindranath Tagore and Sister Nivedita (Bagchi 1990, pp. 68–69). Furthermore, the icon of a "mother India" was also conflated with other Indian goddesses. For example, the figure of the Hindu goddess Kali, depicted in Taikan's painting, was used as a political metaphor in Sister Nivedita's famous book *Kali the Mother*, published in 1900. Vehemently defending the practice of Kali worship in India, often seen as barbaric and superstitious by Westerners, Nivedita's book celebrated Kali as a symbol of Indian motherhood and cultural values (Inaga 2009, pp. 134–36). Through his interactions with Sister Nivedita while in India, Okakura also expressed interest in Kali, as evidenced by his short dedication to the goddess:

> India worships thee in Kali, dread mother of relentless mercy; Japan worships thee in Fudo, grand vision of unflinching pity […] Sleep on, for the hand of Kali shall awaken thee to gleam as gleam the teeth of lightening when the storm laughs on the clouds. Om to the Strong! Om to the Invincible!. (Okakura 1984, p. 166)

Not only did Okakura praise Kali as a powerful symbol, but he also compared the goddess and the Japanese Buddhist deity Fudō Myōō 不動明王. This comparison highlighted his eagerness to culturally unite India and Japan for his Pan-Asianist vision. Thus, more than an exotic representation, Taikan's painting of Kali embodied strong cultural values shared by Indian intellectuals and Okakura. Furthermore, a few years after Taikan and Shunsō's time in India, Abanindranath, following in the footsteps of Chaudhurani and Sister Nivedita, produced the painting *Bhārat Mātā* (Figure 8), or "Mother India," one of the most significant nationalist icons of modern India. Abanindranath's figure of Mother India resembles Taikan's depiction of Kali, with her four limbs, upright posture, and tranquil expression. The painting also utilized the *morotai* style of Taikan and Shunsō in its execution of the background, shown through the fading colors and translucent nature (Satō 1998, p. 89). Such visual characteristics in Abanindranath's painting demonstrated how Indian artists like Abanindranath valued Japanese contributions in crafting their idealized nationalist female icon. From Taikan's

*Indian Guardian* to Abanindranath's *Bhārat Mātā*, we can see how Indian female figures acted as a platform between modern Japanese and Indian artists to experiment and exchange ideas for creating symbolic icons of Indian cultural nationalism.

Naturally, Taikan and Shunsō's exploration of Indian female bodies as powerful icons also made their way into their experimental Buddhist paintings after they returned to Japan. Answering Okakura's call for unconventional Buddhist paintings, Taikan produced two religious-themed works after returning to Japan: *Kannon in White* (Figure 9*)* and *Floating Lanterns* (Figure 10). In the first painting, the artist integrated his observation of Indian female bodies into the figure of the Buddhist deity Kannon, with her large almond-shaped eyes and brown skin. Furthermore, Kannon's body also appeared rotund and fleshy with large thighs and folds of flesh on her neck. She wore a transparent *sari* with her left breast slightly exposed, expressing hints of eroticism.

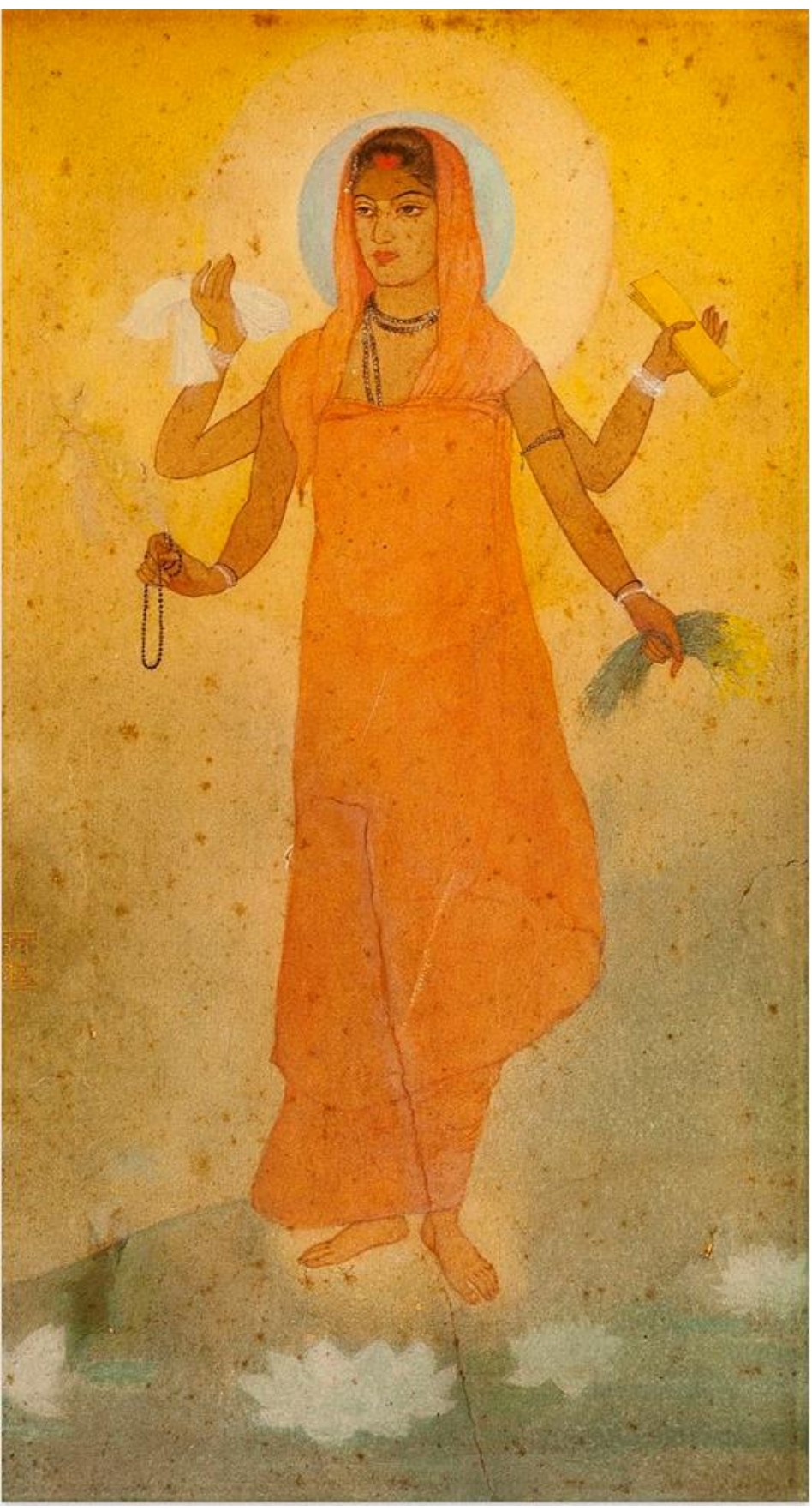

**Figure 8.** Abanindranath Tagore, *Mother India* (*Bhārat Mātā*), 1905, watercolor, 10.5 x 6 in. (26.6 x 15.2 cm). Victoria Memorial, Kolkata, India. Image provided by the museum.

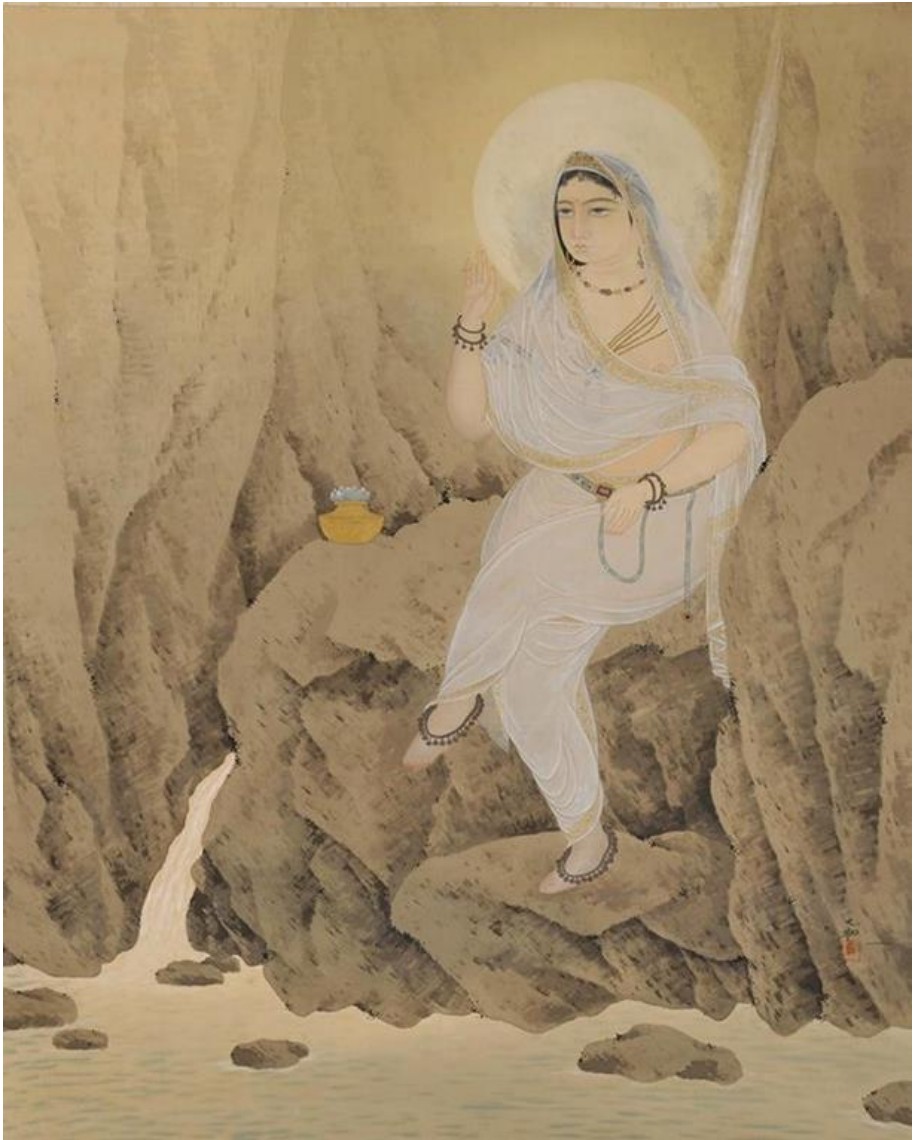

**Figure 9.** Yokoyama Taikan, *Kannon in White* (*Byakue Kannon* 白衣観音), 1908, ink and color on silk, 55.2 x 44.6 in. (140.3 x 113.4 cm). National Museum of Modern Art. Image provided by the museum.

As with *Ras Lila*, Taikan's conception of an Indian Kannon likely came from his observations of Indian women. This is suggested by an audacious comment he made in 1903, where he compared Indian and Japanese body types:

> The [Indian] ladies are even more beautiful and gentle than Japanese ladies. Their facial features are especially wonderful with the very same type of expression as bodhisattvas. A high-class lady is particularly so. With her body covered in glittering gold jewelry and emanating a shining aura, she looks exactly like a painting of Kannon.[7]

The way that Taikan compared an Indian woman to "a painting of Kannon" strongly indicates his conceptual origin of *Kannon in White*. In addition, the ways that Taikan described Indian women with Buddhist similes such as "bodhisattvas" or "shining aura" showed how he fantasized about Indian females as less modern but more religious people and evinced the widespread stereotype of India as a spiritual place in Japan.

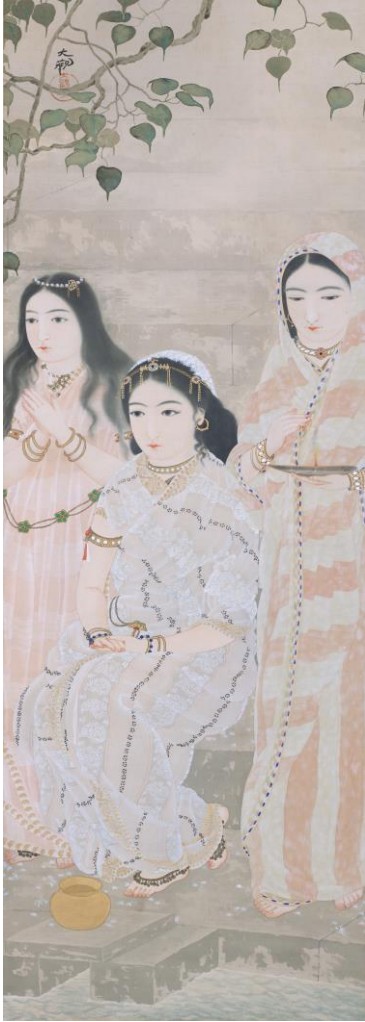

**Figure 10.** Yokoyama Taikan, *Floating Lanterns* (*Ryūtō* 流燈), 1909, ink and color on silk, 56.2 x 20.3 in. (143 x 51.5 cm). Ibaraki Prefectural Art Museum. Image provided by the museum.

While *Kannon in White* fully displayed Taikan's fantasies, *Floating Lanterns* appeared more subtle. Displayed at Japan's Third National Exhibition in 1909, the scroll painting featured three Indian women standing near the banks of the Ganges as they placed floating lanterns down the river as part of a religious ceremony. The painting's inspiration came from Taikan's first-hand observation of a Diwali Festival at Varanasi, where he described young women gathering near the banks of the Ganges to set afloat earthen lamps (Yokoyama Taikan 1968, p. 58). While Diwali is a Hindu festival, Taikan conflated it with the Japanese Buddhist festival of Obon (お盆), where floating lanterns also play a critical role. When the painting was displayed at the Third National Exhibition, critics commended it for both its technical and sentimental achievements and viewers were amused to find that India has a lantern festival similar to Japan.

As Miriam Wattles described, Taikan achieved a sense of "affectivity" as Japanese audiences felt a spiritual connection to India through the painting (Wattles 1996, p. 54). What contributed greatly to this ambiance of affectivity were the figures within Taikan's painting, where the artist incorporated both Japanese and Indian characteristics to construct his ideal image of a spiritual woman that conveyed not only a sense of exoticism but also familiarity to Japanese audiences. Departing from Japanese visual conventions of small, hooked noses and single-lined eyes, the figures' faces instead possess almond-shaped eyes and prominent noses, making the foreign and exotic identities of the women immediately apparent to Japanese audiences (Ibid, p. 53). Yet, Taikan also made small

touches to subdue the Indian-ness of the figures and make them fit more closely to Japanese standards of beauty, such as giving them pale skins and small mouths. Whether it is in *Kannon in White* or *Floating Lanterns*, the portrayal of Indian people in Taikan's paintings represents a physical manifestation of Japan's nostalgic Orientalism.

From the early Indian-themed works of both Taikan and Shunsō, we see how they experimented with Indian female bodies in creative ways. Initially, such figures were developed at the request of Indian intellectuals who collaborated with Japanese artists to create a national female icon. After returning to Japan, however, Taikan utilized what he learned from India to incorporate Indian female figures into Buddhist-themed works that embodied his fantasies of a spiritual yet exotic India. Yet, in works such as Shunsō's *Benzaiten* or Taikan's *Floating Lanterns*, we also see them combine Indian and Japanese traits to create ethnically ambiguous female figures that can serve as transnational icons. Both cases showcased the importance of Indian female figures as a platform for expression for Japanese artists. However, during their time in India, Japanese artists, such as Taikan and Shunsō, fixated on more than just Indian female bodies, for the figure of Śākyamuni also became a coveted icon as artists explored his Indian identity. In their endeavor to study Śākyamuni's depiction in Indian art, Japanese artists flocked to one of the most crucial sources of ancient Buddhist art in India: the Ajanta Caves.

### 5. Creating Śākyamuni—Taikan and Shunsō's Visit to Ajanta

While Taikan and Shunsō spent their time in India studying how to depict Indian female bodies, they also explored visual representations of Śākyamuni in Indian art. As mentioned above, Okakura called for his students to rejuvenate the genre of Buddhist art by taking inspiration from the life of Buddha. However, in addition to the biography of Śākyamuni, Okakura also encouraged Japanese artists to look towards ancient Buddhist art in India for inspiration, and the site that he emphasized most heavily was the Ajanta Caves. Located in modern-day Maharashtra, Ajanta is an ancient cave temple complex dating between the 2nd century BCE and 480 CE, renowned for its wall paintings, which have been touted as one of the oldest Buddhist arts in the world. They were rediscovered by British colonial forces in India during the 19th century and received attention as a crucial archaeological site with both Indian and English artists documenting the cave paintings through copies. Meanwhile, the first mention of the caves in a Japanese publication did not come until 1901 with the manuscript *Abbreviated History of Japanese Imperial Arts* (*Kōhon Nihon Teikoku bijutsu ryakushi* 皇本日本帝国美術略史), where the editorial supervisor, Kuki Shūichi, pointed out how the ancient Indian paintings bear a resemblance to the ancient Buddhist murals at Hōryūji 法隆寺, one of Japan's most important national heritage sites. However, even before the book's publication, Okakura promoted the idea as early as 1890 when he gave lectures at the Tokyo University of the Arts suggesting a connection between the Ajanta Caves and the Hōryūji murals (Fukuyama 2021, pp. 189–90). Thus, it came as no surprise that Taikan and Shunsō, as Okakura's brightest students, would travel to Ajanta to study its cave paintings, mimicking these archaic figures in their art to create an authentic presentation of ancient Buddhist figures.

In Taikan's case, he used Ajanta as a reference to construct his image of Śākyamuni, best evidenced in his now-lost 1903 painting *Śākyamuni Encounters his Father* (*Shaka chi chi ni au*, Figure 11), displayed at the 15th Exhibition organized by the Competitive Painting Association (Kaiga Kyōshinkai 絵画共進会) organized by the Japanese Painting Association (Ibarakiken Yokoyama Taikan Denki Hensan Iinkai 1959, pp. 50–51). As Satō Shino has argued, Taikan modeled several elements of his painting on various images from Ajanta. For example, the appearance of Śākyamuni bears a resemblance to a similar figure of the Buddha from Ajanta's Cave 17 (Figure 12), with his right hand holding an alms bowl and his left hand raised, as with Taikan's depiction. Other aspects of Taikan's paintings incorporated attributes from Ajanta in more subtle ways, such as how the floral

and geometric patterns on the father's belt (Figure 13) were seemingly inspired by similar square patterns from ceiling paintings at Ajanta (Figure 14) (Satō 2010, p. 9).

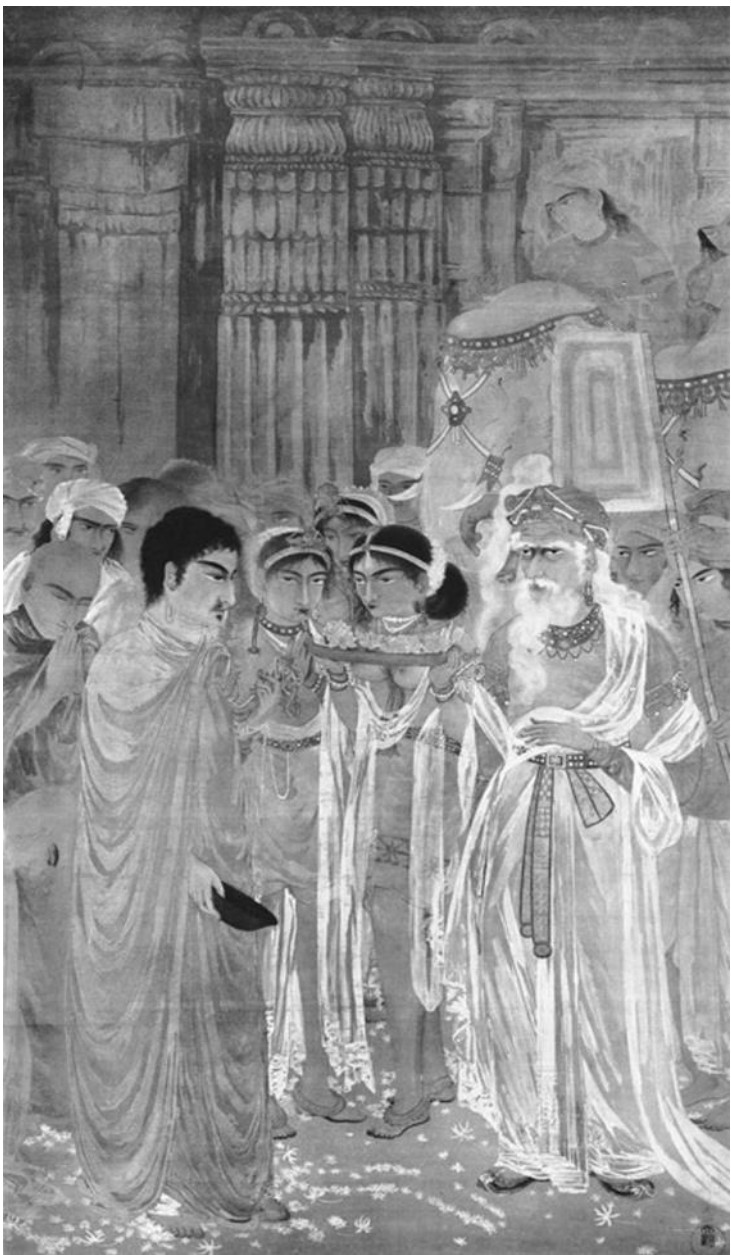

**Figure 11.** Yokoyama Taikan 横山大観, *Sākyamuni Encounters his Father* (*Shaka chichi ni au* 釈迦父に会う), 1903, Photographic reproduction, dimensions unknown. Image: Public Domain.



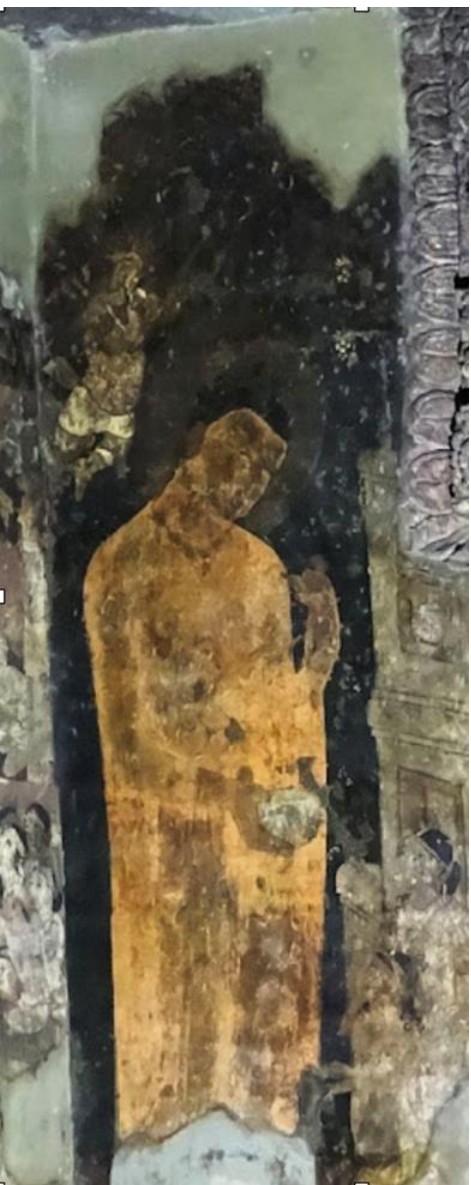

**Figure 12.** Standing Buddha with alms bowl from Ajanta Cave 17, 2nd century B.C–6th century A.D, mineral pigments on plaster, dimensions unknown. Photograph by Benoy K Behl. Image: Courtesy of the photographer.

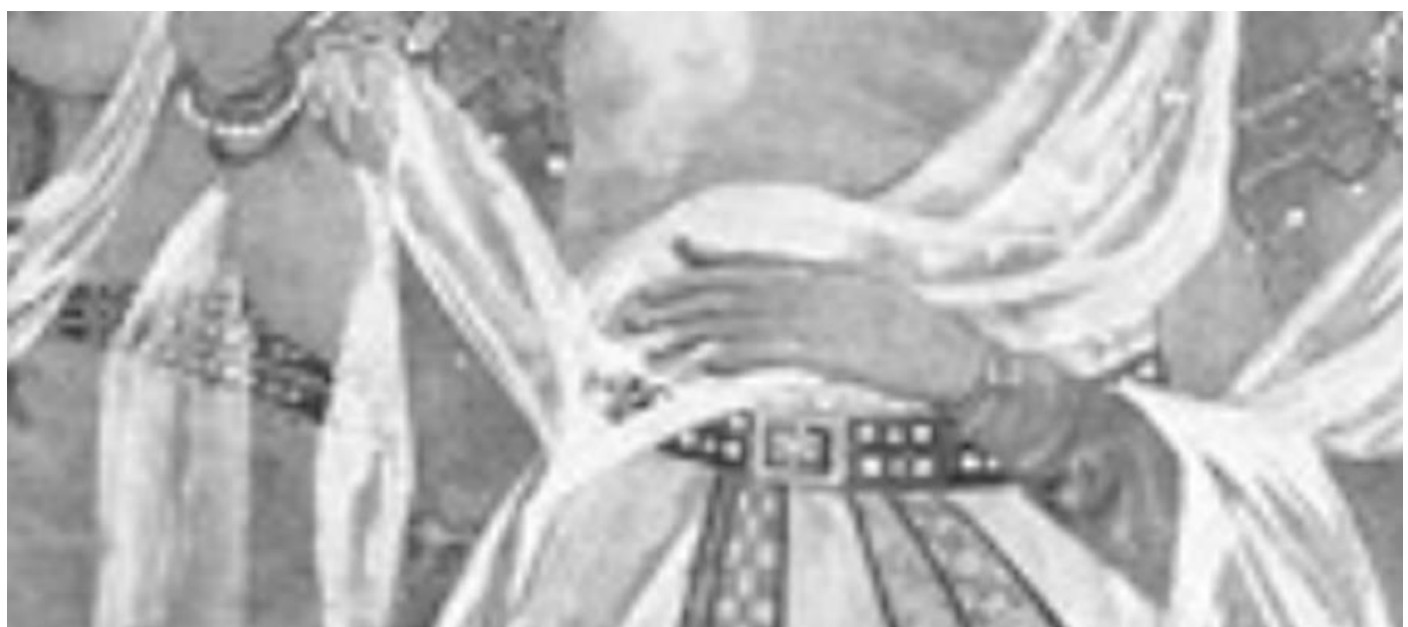

**Figure 13.** *Sākyamuni Encounters his Father*, details.

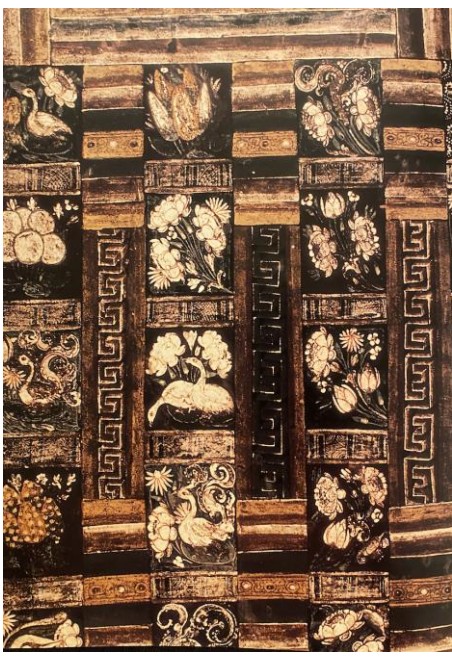

**Figure 14.** Ceiling painting from Ajanta Cave 17, 2nd century B.C–6th century A.D, mineral pigments on plaster, dimensions unknown. Notice the grid pattern, similar to the belt in *Sākyamuni Encounters his Father* (see Figure 13). Photograph by Benoy K Behl. Image: Courtesy of the photographer.

When the work was displayed in Japan, reviewers praised Taikan's authentic portrayals of Indian people, describing the viewing experience as comparable to "viewing a scene from ancient India."[8] One review noted that "the figures appear to be sketched from real-life Indians, but their postures are strangely missing, and the painting avoided giving a sense of unnaturalness."[9] Such reviews came as no surprise, for even though only photographic reproductions of this painting remain today, viewers can see exotic Indian features on both figures, such as Śākyamuni's mustache and his father's long beard and turban (Auerback 2016, p. 218). As these reviews showed, Japanese audiences valued modern Buddhist paintings for their portrayals of Indian people, and racial authenticity was a crucial factor in how they judged the figures that appeared in such paintings.

However, as the latter quote showed, they also looked upon the Indian people with an exotic gaze, expecting them to give off a sense of "unnaturalness" or having certain postures. These responses reflected the mixed attitudes that Japan had towards India, where they admired the nation as the birthplace of Buddhism yet objectified its people as exotic foreigners.

Taikan's work represented an early example of *nihonga* artists using the Ajanta Caves as a reference to create a historical and Indian Śākyamuni faithful to the roots of Buddhism. However, with the arrival of the next Japanese artist in India after Taikan and Shunsō, Katsuta Shōkin (1879–1963), we see more diverse approaches in capturing and interpreting Śākyamuni in painting.

### 6. Creating Śākyamuni—Katsuta Shōkin and His Portraitures

After Okakura's two students returned to Japan, Rabindranath Tagore eagerly asked him to send another artist to teach his family and his school. Thus, in 1905, Okakura chose a young graduate from the Tokyo School of Fine Arts named Katsuta Shōkin for this undertaking. Today, Shōkin is largely remembered as a respected bird-and-flower artist, while his activities in India and related works receive little attention. That is because nearly all of his Indian-themed works remain missing, leaving only rough sketches or black-and-white reproductions of his works to offer us a glimpse. Despite this, even in what remained of Shōkin's time in India, we can see the artist's passion for exploring Indian bodies and ethnicity in Buddhist paintings. However, unlike Taikan and Shunsō who explored exotic femininity in Indian bodies, Shōkin spent more time exploring the "Indian-ness" of the historical Buddha.

The only Indian-themed painting by Shōkin that can be viewed today is his 1907 work, *Śākyamuni's Departure* (*Shūtsujō Shaka*, Figure 15), which depicted the young Śākyamuni wearing royal garments, standing near his palace entrance and next to a servant with a horse. The painting depicts an episode from Śākyamuni's life when the then-young prince made the momentous choice to leave his luxurious palace and family to pursue an ascetic life, marking the start of his path to enlightenment. In her analysis of the painting, Narihara Yuki highlights how Shōkin incorporated both his observations of Indian people and ancient figures from Ajanta to create a historically and ethnically authentic Śākyamuni in his portraiture. For example, the prince's pose in the painting closely resembled the pose of a figure (Figure 16), possibly a monk, that Shōkin sketched in his notebook from his time in India (Narihara 2005, p. 553). Both figures stood with their weight on their left foot while holding up their clothes with their left arm. What set Śākyamuni apart from the sketched figure was that he stood confidently taller and wore more jewelry and adornments, conveying his royal status. However, the biggest difference lay in the face and crown of Śākyamuni, which Narihara argued was inspired by a figure from Ajanta (Figure 17).

If we compare the head of Shōkin's Śākyamuni (Figure 18) with the figure from Ajanta, we can see several resemblances. Both Shōkin's and Ajanta's figures feature a dark-skinned male with long slender eyes and long black hair matted onto the shoulders. The elaborate crown that Śākyamuni wore in Shōkin's painting also resembled the one worn by the Buddha figure from Ajanta, with strings of pearls hanging from it (Narihara, p. 557). Just as Taikan and Shunsō referenced contemporary Indian art and observations of Indian locals to construct their figures, Shōkin looked towards both Ajanta's ancient paintings and the body of an Indian man to create his idealized version of the young Śākyamuni. In doing so, he portrayed the Buddha, not as the cosmic entity worshipped in Japan for many years, but as a historical figure.

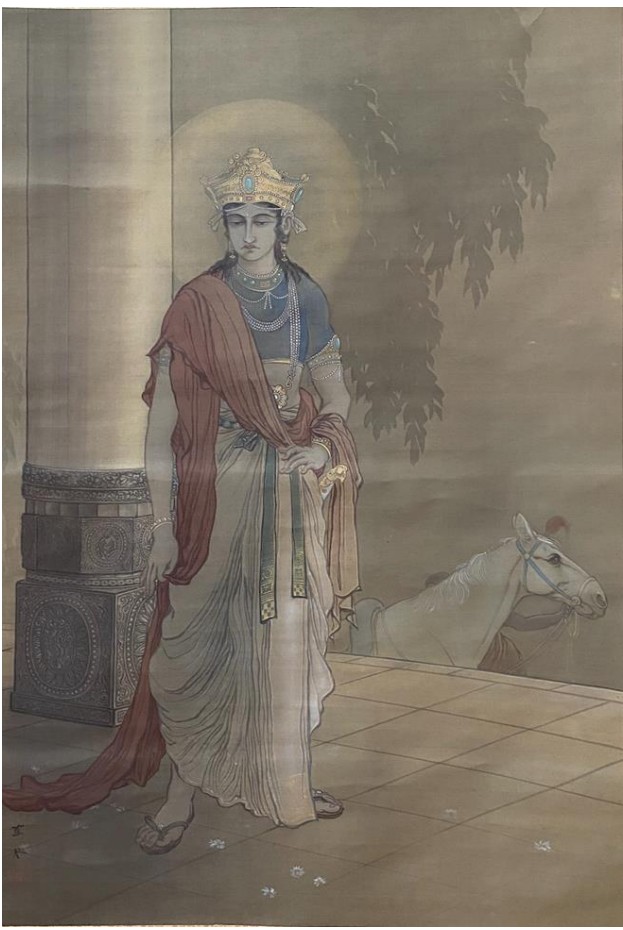

**Figure 15.** Katsuta Shōkin 勝田蕉琴, *Sākyamuni's Departure* (*Shutsujō Shaka* 出城釈迦), 1907, ink and color on silk, 38.7 x 26.3 in. (98.4 x 66.8 cm). Fukushima Prefectural Art Museum. Image provided by the museum.

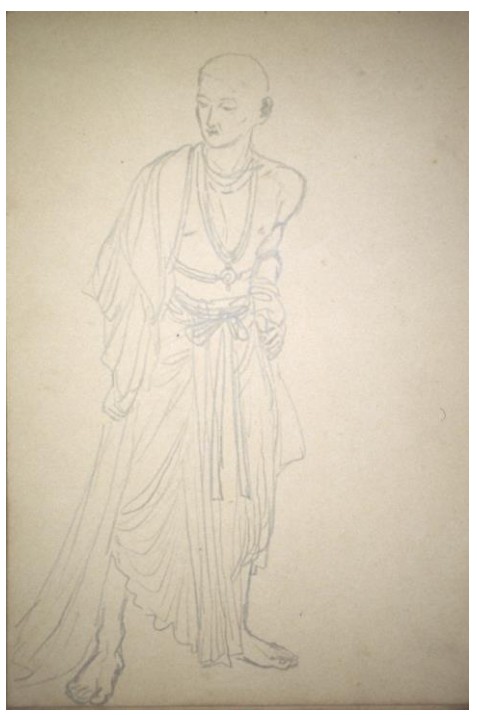

**Figure 16.** Katsuta Shōkin, untitled sketch of a monk figure, date unknown (circa 1905-1906), pencil on paper, dimensions unknown. Tokyo University of Art. Image provided by the university.

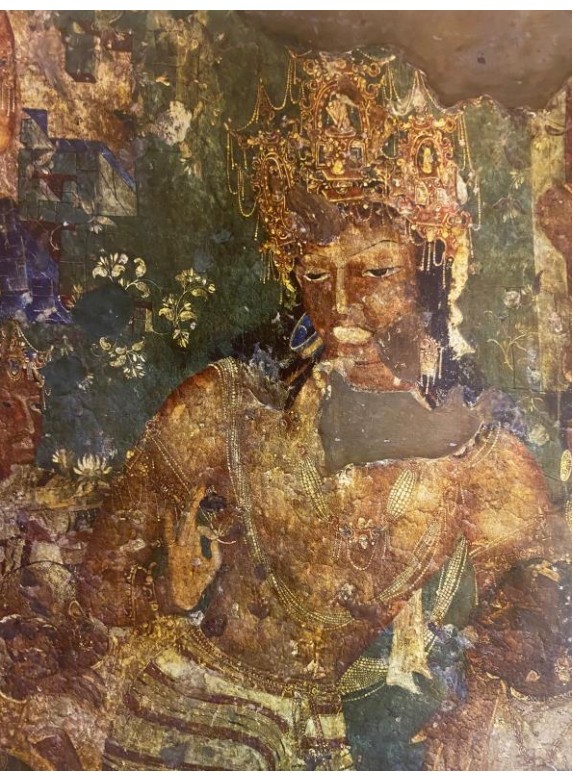

**Figure 17.** Bodhisattva Vajrapani figure from Ajanta Cave 1, 2nd century B.C–6th century A.D, mineral pigments on plaster, 7 x 3 ft. (2 x 1 meters). Photograph by Benoy K Behl. Image: Courtesy of the photographer.

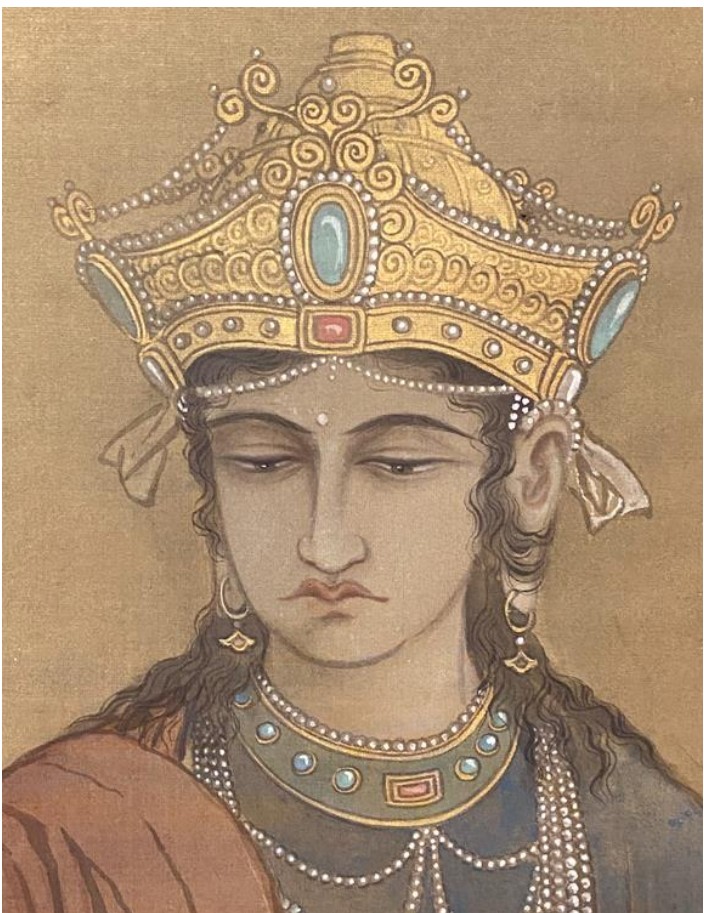

**Figure 18.** *Sākyamuni's Departure*, details. Notice the similarities with the figure of Bodhisattva Vajrapani from Ajanta, particularly the crown and long hair.

However, although Narihara's analysis showed how Shōkin's portraiture of Śākyamuni referenced many of his observations while in India, a little-known colored sketch (Figure 19) by Shōkin that also illustrated the story of Śākyamuni's departure showed how the artist attempted a different approach in portraying Śākyamuni. It is unknown as to whether Katsuta drew this sketch as the original conception for *Śākyamuni's Departure* or as a standalone piece. Regardless, similarities in the pose and color scheme of Śākyamuni's figure from both works suggest a strong relationship. Like the painting, the colored sketch showed Śākyamuni alone, accompanied only by his horse. Unlike the painting, though, he was surrounded by tropical vegetation, and his attire differed quite drastically. Instead of an elaborate crown and jewelry, Śākyamuni's robe offered few visual clues to his royal status save for a diadem, necklaces, and a sword. Most importantly, his right shoulder and chest were laid bare and he wore no shoes, making him closer in resemblance to the sketched monk figure from Shōkin's notebook. The whole composition portrayed Śākyamuni closer to his persona as a forest ascetic than as a royal prince. Thus, it is plausible that this sketch was narratively the sequel to Śākyamuni's departure where the prince reached the forests and discarded his royal attire.

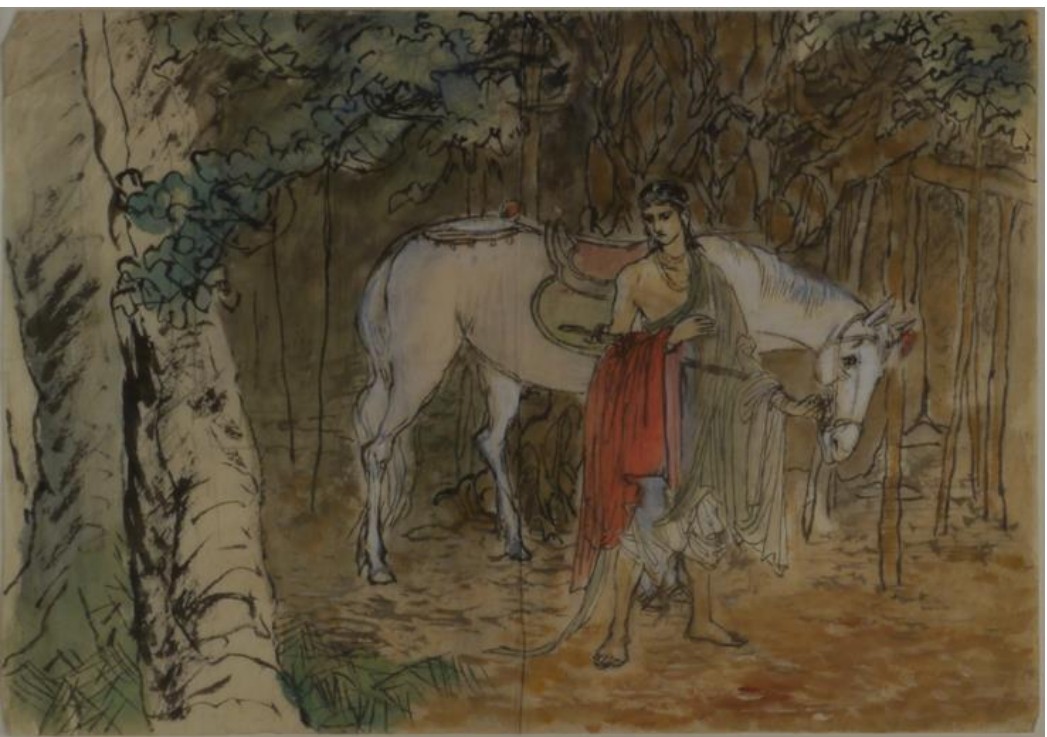

**Figure 19.** Katsuta Shōkin, untitled colored sketch of Śākyamuni in the forest, date unknown (circa 1905-1906), ink and color on paper, dimensions unknown. Fukushima Prefectural Museum of Art. Image provided by the museum.

Compared to *Śākyamuni's Departure*, the colored sketch expressed a more exotic atmosphere with Śākyamuni barefooted and half-dressed while surrounded by tropical vegetation. Unlike Shōkin's finished painting, which referenced India's ancient history with the stone pillar in the background and Śākyamuni's attire, the colored sketch made more references to Japanese imaginations of contemporary India as a tropical country of people dressed in light clothing. It also evoked imaginations of ancient civilizations as Japanese contemporaneous scholars frequently compared ancient India to other archaic civilizations such as ancient Greece and Egypt. It revealed how even though Japanese artists aimed to authentically depict Śākyamuni closer to his Indian roots, they were also eager to use the figure of Śākyamuni similar to Indian female figures: as vessels for their exotic fantasies.

Following *Śākyamuni's Departure*, Shōkin painted another version of Śākyamuni in the now-lost painting titled *Prince Siddhārtha beneath the Jambu Tree* (*Enbujuka no Shidda Taishi*, Figure 20) in 1908. This work portrayed him more as a religious figure than a historical prince. Although only photographic reproductions remain today, viewers can see how the painting diverged from *Śākyamuni's Departure*. In the depiction, Shōkin abandoned rich jewelry and a confident posture in favor of a more reserved Śākyamuni, with his head bowed and hands clasped together at the front near his waist. Instead of a palace entrance, the background featured the titular Jambu tree near a stone platform. With this depiction, Shōkin aimed to create a sentimental painting that highlighted Śākyamuni's spiritual devotion. However, when the painting was displayed at the first National Painting Achievement Association's (Kokuga Gyokuseikai 国画玉成会) exhibition, which showcased various avant-garde Japanese-styled paintings by young artists, it received some harsh criticisms. One account mocked how Śākyamuni looked like "a urinating young prince."[10] Another criticism pointed out how Shōkin's new version of Śākyamuni "appeared more like a foreigner [Westerner], and not like an Indian."[11]



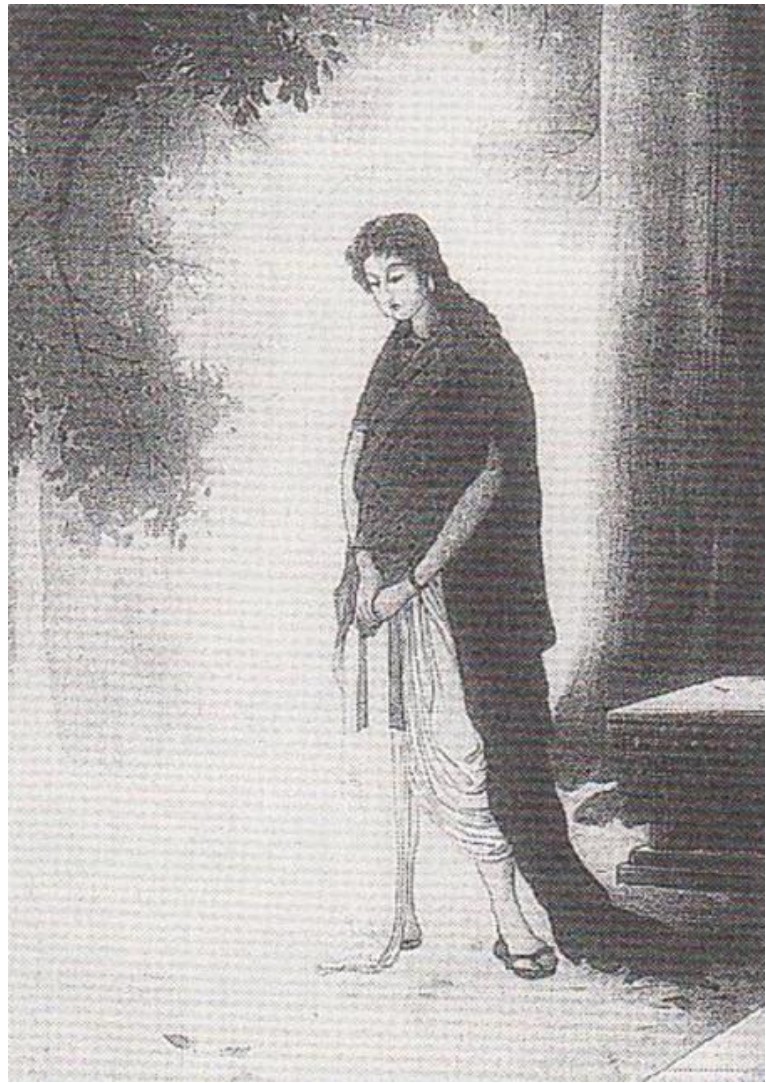

**Figure 20.** Katsuta Shōkin, *Prince Siddhārtha beneath the Jambu Tree* (*Enbu juka no Shitta Taishi* 閻浮樹下の悉達太子), 1908. Fukushima Prefectural Museum of Art. Image provided by the museum.

These criticisms offered interesting insights into the reception of Śākyamuni in Japanese art, when audiences had come to expect historical Buddhist figures to appear Indian. However, it can also be argued that they were simply looking for exoticism rather than spirituality in the Indianized figure of Śākyamuni, which reflected the mixed attitude that artists, including Shōkin, and viewers had towards India. Undeniably, Shōkin dedicated much effort to constructing an authentic figure through his studies of Ajanta and Indian locals. However, he also imbued his art with his fantasies of India, with the tropical scenery and half-dressed monk figures. Similar to his predecessors, Shōkin's adaptation of India into art is characterized by an admiration for its innate spirituality and cultural ties with Japan, but also a condescending objectification of its people.

## 7. Conclusions

The early twentieth century was a time of transformation for Buddhist understanding in Japan where scholars and artists started to recognize the importance of India as the birthplace of Buddhism. Western academicism helped Japanese scholars and artists gain a new appreciation for the country, which motivated artists to travel to India. In India, these artists searched for inspiration for a new kind of Buddhist art that visualizes the religion's Indian connection, and what immediately attracted them were the bodies of Indian locals. Artists such as Taikan and Shunsō looked toward the

depictions of Indian women as new icons for Buddhist art, combining them with Japanese traditional portrayals of women to create culturally fluid figures in their quest to create idealized oriental bodies. Others, such as Shōkin, explored the Indian ethnicity of Śākyamuni, referencing the Ajanta Caves and Indian locals to depict him authentically. Whether it was an attraction to the imaginations of Indian women or to the figure of Sākyamuni, Indian bodies served as a figural prototype for modern Buddhist art in Japan.

The artists' desire to travel to India for artistic inspiration showcased their respect for India as a repository of Buddhist heritage. However, they also adopted an objectifying and voyeuristic gaze and framed Indian culture and people in an exotic light. Indo–Japanese relationships during the twentieth century cannot be defined easily and modern Buddhist works produced by Japanese artists who traveled to India reflected this complexity. Showcasing wide varieties of approaches and innovations in adapting India into their works, especially as human figures, these Japanese artists showed how cross-cultural exchanges could lead to unexpected developments in Buddhist art.

**Funding:** This research received no external funding.

**Data Availability Statement:** No new data were created in this research. Data availability is not applicable to this article.

**Conflicts of Interest:** The author declares no conflicts of interest.

## Notes

1. For a detailed explanation on the *trikāya* and Śākyamuni's place in Mahāyāna Buddhism, please also consult Judith Snodgrass's work (Snodgrass 2012, pp. 87-90).
2. For more information on early Japanese pilgrims to India and their effects on future travelers, please read Chapter 1 of Richard Jaffe, *South Asian Encounters: Kitabatake Dōryū, Shaku Kōzen, Shaku Sōen, and the First Generation of Japanese Buddhists in South Asia*.
3. Okakura Kakuzō posted the "Call for Submission: Buddhist Painting Prize" ("Kenshō butsuga boshū" 懸賞仏画募集) on volume 105 of *Kaiga sōshi* 絵画叢誌 (page 3), which was published on October 25, 1895. The announcement is translated and cited in Auerbach 2016, pp. 204–205.
4. For more details on Okakura's Buddhist painting competition, please also see Nakano Shinsuke's work (Nakano 2016, pp. 357 to 359).
5. Translated by Micah Auerback. I referenced the excerpt quoted in *Nihon Bijutsuin hyakunenshi* (Nihon Bijutsuin Hyakunenshi Henshūshitsu 1989, p. 530).
6. Ibid.
7. "Taikan and Shunsō Talk India in Kyoto" ("Taikan, Shunsō Kyōto de no Indo-dan" 大観、春草京都でのインド談). I reference the version reprinted in Yokoyama Taikan Kinenkan, ed., *Taikan no garon*, 43, which was then cited and translated by Miriam Wattles in her article (Wattles 1996, p. 54).
8. Kaiga Kyōshinkai Shinbun hyō" 絵画共進会新聞評 included in volume 58 of *Nihon bijutsu* 日本美術, published in 1903. My observation is based on the excerpt Satō Dōshin quoted in his study (Satō 2010, p. 10).
9. Ibid.
10. Published in *Kokumin shinbun* 国民新聞 issued in 1908 (month and day unknown). I referenced the excerpt reprinted in *Nihon Bijutsuin hyakunenshi* (Nihon Bijutsuin Hyakunenshi Henshūshitsu 1989, p. 753).
11. Appeared in volume 118 of *Nihon bijutsu*, published in 1908. I referenced the excerpt reprinted in *Nihon Bijutsuin hyakunenshi* (Nihon Bijutsuin 1992, p. 753).

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
