# Peer review of "‘Bodhisattva Bodies’: Early Twentieth Century Indian Influences on Modern Japanese Buddhist Art"

_arts, 2023_

Round 1

Reviewer 1 Report

Comments and Suggestions for Authors

Reconsider the title "Between Body and Spirit." A) this sets up a very European false dualism between visible bodies and invisible religious ideas. All religious art is visible; it’s just that Japanese artists borrowed Indian visual codes during this period.  B) it is unclear what the author means by the rest of the title. Is India supposed to be associated with Body and Japan associated with Spirit? Or vice versa? Or just that Japanese "spiritual" art shows both Indian and Japanese figures and physical features? C) the term "spiritual" is never defined (and indeed can never be defined because it has become such a vague referent in popular speech) D) it is just very distracting and unnecessary. Either find a better precursor before the colon or just cut it altogether.

p. 1 line 22 – where did this 1902 meeting take place? In India or Japan? P. 6 lines 234-235 says that Okakura went to India to invite him to Japan, so I assume the author means the first meeting occurred in India. But Vivekananda had already visited Japan en route to the US in 1893. Why isn’t this cited as the first cultural exchange between the two countries? Or all the others in Richard Jaffe’s book? Perhaps it would be better to specify that although there were many preceding religious and scholarly exchanges between India and Japan, arguably the most significant or impactful artistic exchange occurred in 1902.

p. 3 Please revise this paragraph on Mahāyāna Buddhism.

·       line 100 – Mahāyāna proposes the three body (trikāya) doctrine that recognizes the historic Buddha-body (nirmanakāya) - exemplified by Shakason, heavenly/deified Buddha body (sambhogakāya) – exemplified by Amida butsu, and cosmic world-body of Buddhahood (dharmakāya) – exemplified by Dainichi nyorai.

·       lines 101-103: This citation from Auerbach is problematic. Even the earliest Buddhas that entered Japan exhibit a range of identities; not just seated Buddhas with hands in meditation pose (zenjō). There are Sakyamuni figures (7th c. Hōryuji in meditative pose, yes, but also in teaching mudra); Amida (7th c. Tachibana shrine; Hōryuji western wall fresco, Sarusawa Pond Amidaji in Nara); Yakushi nyorai (Yakushiji; Shinyakushiji; Ganjin’s standing Yakushi at Tōshōdaiji, or the early Heian standing Yakushi at Jingōji in western Kyoto / Arashiyama), and Vairocana / Birushana (8th c. Tōdaiji  - the dharmakāya in boon-bestowing varada-mudrā & and fear-not abhaya-mudrā). Then the pantheon explodes with Kūkai’s introduction of mikkyō in the 9th c.  

p. 3 paragraph on Meiji modernism

·       The next paragraph makes a BIG historic leap from the previous one (7th c. to 19th c.). I would honestly rewrite that whole introductory Mahāyāna paragraph and make it flow more smoothly into the next one somehow.

·       Line 108 – “Western academicism” is very vague. It would be more helpful to specify that the so-called “scientific” discipline of Buddhist Studies in Japan was specifically modeled on the new 19th c. textual-historical method in Biblical Studies, and that the late 19th/early 20th c. “recovery” of Buddha’s biography in Lumbini, Bodhgaya etc. mirrored and was influenced by the archaeological ‘search for the historical Jesus.’

·       In addition, I think a sentence or two more is needed – to explain how Japanese Buddhist clerics, scholars, and artists of the time wanted to mine the Indian past precisely in order to fashion a new, modern future for Japanese Buddhism. By setting up Japan as the farthest and furthest culmination of classical Indian Buddhism, and by simultaneously integrating the latest Euro-American political-economic and intellectual-cultural developments, Japan positioned itself as the leader of a self-styled pan-Asian Buddhasphere that was uniquely poised to resist Euro-American imperialism on its own terms and beat them at their own game.

p. 3 – it is usually referred to as the World Parliament of Religions. Revise throughout for the sake of consistency. Also – be careful. The author makes it sound as if it were the Japanese delegates at the WPoR (i.e. Shaku Sōen and DT Suzuki) who denigraded Dharmapāla’s Theravada tradition – when in fact Shaku Sōen himself had studied abroad in Sri Lanka and greatly respected his teachers there. Be more precise in saying that there was a range of Japanese views about Theravada at the time, but that it was other Japanese clerics and scholars back home in Japan who tended to voice patronizing views of their Buddhist cousins in Sri Lanka.

p. 4 line 170 – “the cosmic Buddha” refers to Mahāvairocana / Dainichi – the principal personification of the dharmakāya / universal Buddhahood in the Taimitsu (Tendai) and tōmistu (Shingon) strains of esoteric Buddhism in Japan (mikkyō). But Amida, Yakushi, Sakyamuni Buddha continued to be venerated along with all the popular bodhisattvas, wisdom kings, devas (-ten) and gongen for centuries.

p. 5 – line 189 – give full names the first time you mention Yokoyama and Hishida. Also – include all the relevant figure numbers after you first mention their morotai “vague” style – so that we can see what morotai means and looks like right away.

p. 5 line 212 – this is not a conventional arrangement for the Buddha’s birth scene. It’s usually shown as Queen Māyā holding onto the sal trees in labor with Buddha emerging from her side, attended by Mahāprajāpati and Indra as the babycatcher, with the Buddha then taking 7 steps on 7 lotus flowers and pointing above and below to proclaim that he will be ruler of all. It would be better to note in this scene how Shimomura Kanzan deliberately drapes one of the Buddha’s disciples (anachronistically) in Persian textiles - that feature the characteristic blue medallion designs - that are found in Emperor Shōmu’s 8th c. Shōsōin warehouse.

p. 9 lines 381-382: fascinating. This complicates the narrative of Japan paradoxically colonizing Asia to save it from the colonizers. It demonstrates that Japan didn’t always engage in cultural appropriation. India was a willing partner – and indeed commissioned outsiders to reinterpret their own artistic traditions for themselves; and Tagore actively invited Japanese artists to come teach them Nihonga ptg.

p. 10 lines 430-431: I believe the idea that India is spiritually superior to the West derives from Vivekananda and other Hindu nationalists. It is this notion that Japan mimics.

p. 11 line 455 – & p. 12 line 517 – you should coin the phrase Japan’s “nostalgic Orientalism” for India.

Interesting contribution to this growing field of scholarship! Thanks for the opportunity to review.

Recommend publication with revisions.

Comments on the Quality of English Language

A good copyeditor should be able to catch some of the missing words and typos.

Author Response

Dear Reviewer

Please see attached Word file for my reply

Sincerely,

Reviewer 2 Report

Comments and Suggestions for Authors

This is a very well written and fascinating paper, thank you! My only suggestion is to use diacritics consistently when presenting Sanskrit terms. They are used sporadically, and should be either removed entirely or use consistently and correctly. If you are interested in the latter option, please consider the following changes:

Line 40, 97, 159, 165, 167, 169, 211, 470 et al.

Correct “Sākyamuni” to “Śākyamuni”

Line 127

Correct “Hīnayana” to “Hīnayāna

Line 127

Correct “Mahayana” to “Mahāyāna

Line 129

Correct “Maha” to “mahā

Line 132-133

Correct “Mahayana” to “Mahāyāna

Line 279, 300

Correct “Krishna” to “Kṛṣṇa”

Line 280, 329

Change “Gopis” to “Gopīs

Line 303, 304, 307, 309, 312, 315, 317, 325, 385, 388, 390, 396, 399

Correct “Kali” to “Kālī”

Line 342, 348, 352

Correct “Sarasvati” to “Sarasvatī”

Line 350

Correct “vina” to “vīṇā

Author Response

Dear Reviewer

Thank you so much for your time and your kind words. I have read your suggestion and I agree that I should keep my Sanskrit terms consistent. However, I would like to ask if I can keep the name "Krishna" in English as it is the name that's more familiar to English readers. Otherwise, I'm also fine with using the Sanskrit name "Krsna."

I would greatly appreciate your thoughts on this. Please let me know what you think.

Sincerely,

Reviewer 3 Report

Comments and Suggestions for Authors

See the attachment

Author Response

Please see attached file for my replies. I sincerely thank you for your time and reviews
